# Conflicting Bundle Allocation with Preferences in Weighted Directed Acyclic Graphs: Application to Orbit Slot Allocation Problems †

**Stéphanie Roussel** [1,*] **, Gauthier Picard** [1] **, Cédric Pralet** [1] **and Sara Maqrot** [2]

1 ONERA/DTIS, Université de Toulouse, F-31055 Toulouse, France; gauthier.picard@onera.fr (G.P.); cedric.pralet@onera.fr (C.P.)
2 Berger-Levrault/DRIT, 31670 Labège, France; sara.maqrot@berger-levrault.com
* Correspondence: stephanie.roussel@onera.fr
† This paper is an extended version of our paper published in Advances in Practical Applications of Agents, Multi-Agent Systems, and Complex Systems Simulation. The PAAMS Collection. PAAMS 2022. Lecture Notes in Computer Science; Springer: Cham, Switzerland, 2022; Volume 13616.

**Abstract:** We introduce resource allocation techniques for problems where (i) the agents express requests for obtaining item bundles as compact edge-weighted directed acyclic graphs (each path in such a graph is a bundle whose valuation is the sum of the weights of the traversed edges), and (ii) the agents do not bid on the exact same items but may bid on conflicting items that cannot be both assigned or that require accessing a specific resource with limited capacity. This setting is motivated by real applications such as Earth observation slot allocation, virtual network functions, or multi-agent path finding. We model several directed path allocation problems (vertex-constrained and resource-constrained), investigate several solution methods (qualified as exact or approximate, and utilitarian or fair), and analyze their performances on an orbit slot ownership problem, for realistic requests and constellation configurations.

**Keywords:** path allocation; fairness; constraint optimization; satellite constellation





## 1. Introduction

Earth observation satellites capture a vast number of images of the Earth's surface every day. These images are delivered to end-users who have made observation requests for several purposes such as monitoring critical areas affected by natural disasters or crises, observing infrastructures, monitoring the environment, etc. The observation request process operates in the following manner. First, users submit their observation requests to the main mission center. The mission center then computes observation plans which are transmitted to the satellites when they overfly a ground control station. Subsequently, each satellite captures the requested images and transmits the collected data when it passes over a ground reception station. The satellites we consider in this work are on low Earth orbit and complete around 16 orbits per day, which allows them to pass over several Earth areas at different times every day.

In order to improve the capability to deliver images as early as possible after requests are formulated, one can rely on constellations of Earth observation satellites that are currently deployed. Constellations also offer the possibility for users to express more complex requests. An example of such a complex request is a *periodic request*, that consists in observing an area of interest at regularly spaced dates. Generally, the number of posted requests on a given time horizon is too large to satisfy them all. Therefore, the main mission center has to select which requests to perform for the upcoming time horizon, for instance using manually defined prioritization rules. As such a selection process does not offer guarantees for users with regards to the satisfaction of their requests, Earth observation satellite constellations' managers now propose a new observation paradigm, namely, *exclusivity orbit slots* booking. Whenever users

buy exclusive orbit slots of a satellite, they can exploit this satellite during the associated time windows using their own ground stations. This allows users to send observation plans to satellites and collect the observations realized during the orbit slots.

In this respect, from the point of view of the operator of an Earth observation satellite constellation, we consider the following problem. The goal is to attribute ownership of some orbit portions to several clients. Each client has some points of interest (POIs) to acquire at some frequency, e.g., capture L'Aquila city every 2 h for 6 months. Since several satellites may capture the very same point on Earth around the requested observation times, several possible bundles of orbit slots are specified by each client, together with a preference for some bundles depending on the quality of the sequence of orbit slots, e.g., based on the POI viewing angle provided by each slot. Moreover, as several clients may be interested in very close POIs, several requested orbit slots may overlap. Each orbit slot in this category can be either allocated to a single client or divided between clients. These situations can be captured by the models we propose in this article.

More precisely, we consider a problem of allocation of conflicting bundles of items constrained by item chaining (to allocate to each agent a chain of successive items). The chaining constraint is captured by using, for each agent, an edge-weighted directed acyclic graph (DAG) representing all the valid bundles (i.e., paths) of items for the agent, where the quality of a bundle is represented by additive edge weights. Then, conflicting bundles cannot be allocated at the same time and have to be handled so that each agent obtains one conflict-free path in its graph. Such a setting occurs in application domains such as network function virtualization (NFV), where users request allocating directed graphs of services into a shared networked infrastructure [1]. As explained before, this also occurs in Earth observation using a constellation of satellites in a scenario where users demand the ownership of some repetitive orbit slots, without overlapping with other users' slots, to fulfill periodic observation requests [2,3]. In such settings, beside the additive edge weights, other criteria can be considered to guide the allocation process, especially when constellation users are stakeholders expecting allocations to be fair or proportional to their investment.

In this paper, we contribute on the following points:

- We define a generic modeling framework for the path allocation problem with conflict (directed path allocation problem, or DPAP) and consider two optimization criteria (global utility and leximin).
- We instantiate this framework with two compact representations of conflicts, one based on a vertex conflict (vertex-constrained directed path allocation problem, or V-DPAP) and one based on a resource consumption conflict (resource-constrained directed path allocation problem, or R-DPAP)—note that V-DPAP comes from the path allocation in the directed acyclic graph (PADAG) problem defined in [4].
- We show that the decision problems associated with V-DPAP and R-DPAP are NP-complete, whatever the optimization criteria.
- We define several complete and incomplete allocation schemes for solving V-DPAP and R-DPAP.
- We evaluate all of the algorithmic approaches on dozens of orbit slot allocation benchmarks and discuss the obtained results.

The paper is structured as follows. Section 2 discusses related works focusing on the allocation of goods as paths. Section 3 presents the DPAP framework to tackle path allocation in multiple conflicting edge-weighted directed acyclic graphs. In Sections 4 and 5, we consider vertex-based conflicts (V-DPAP) and resource-based conflicts (R-DPAP). We analyze the theoretical complexity of the associated decision problems and discuss the relationship between the two frameworks. Section 6 lists some algorithms, complete and incomplete, that can be used to solve V-DPAP and R-DPAP. Section 7 presents the experiments used to evaluate the performances and behaviors of our solution methods on problem instances coming from the Earth observation domain. Finally, Section 8 concludes the article with some perspectives.

## 2. Related Works

The literature contains some work related to the allocation of goods structured as graphs. In fair division of graphs, the objective is to divide a graph of items between several agents, with additive utilities attached to nodes [5,6]. These works provide interesting properties to find envy-free or Pareto-optimal, allocations in an efficient manner in some specific graph structures, e.g., paths, trees, stars. However, in our problem, (i) agents do not compete for the very same set of items, (ii) the graph is directed to compose paths from a start time to an end time, and (iii) even by mapping our problem to a graph division problem and by regrouping conflicting items into composite items, it is highly improbable that the resulting graph is acyclic. Here, graphs are used to express preferences and not the goods to allocate. In short, our work does not fall into the existing graph fair division frameworks, and cannot benefit from theoretical results on path-shaped or star-shaped graphs.

Another related method is path auctions [7–9], where agents bid for paths in a graph where each edge is owned by an agent. The goal is to assign paths to agents by the means of auctions, and optionally to keep some privacy for the edge owners. In the case of a utilitarian objective function for the winner determination problem, without price privacy, this falls into the Vickrey–Clarke–Groves framework, and thus guarantees some efficient and *strategy-proof* mechanisms. However, here again, agents bid on the very same set of nodes and edges.

In the transportation domain, investigations on very similar structures, that is flow networks, provide techniques for fair maximum flow in multi-source and multi-sink networks [10]. While the techniques used are very similar to ours (linear programming), the maximum flow objective is very different from path utility maximization with a single path per agent. Furthermore, [11] worked on multiple shortest path problems based on deconflicting techniques. While the problem displays similar characteristics, once again the agents evolve on the very same graphs, and the objective is focused on minimizing path length and minimizing conflicting paths, without fairness desiderata.

In congestion games, agents are allocated paths so that the delays incurred by crossing paths are minimized. The more agents are allocated the same nodes, the more delay is attached to their paths [12,13]. In our work, we do not consider delay but incompatibilities. Even if they could be modeled as nonlinear $\{0, \infty\}$ functions, in our problem some path allocations are unfeasible, contrarily to congestion games. Furthermore, using congestion game solution methods, as in [13], may result in unfair Nash equilibria, because of numerous unfeasible paths[1].

More generally, another classical approach to the fair allocation of indivisible goods is *round-robin*, which is almost envy-free [14]. This is notably one favored technique to allocate virtual network functions in network function virtualization infrastructures [15], or to schedule tasks. We will use it as a competitor for our techniques.

In [16], we proposed constraint-programming approaches for fair sharing of orbit slots in the case of Earth observation satellites. We considered several types of requests, such as periodic and global requests. The latter type of requests cannot be modeled within the graph-based framework proposed in this paper. Therefore, we had to enumerate all the ways to (partially) satisfy requests. This enumeration is not required within the framework we propose here, because of the compact graph representation. Moreover, the approaches of [16] were evaluated on small horizons, due to the computational intensiveness of the proposed solution methods. The horizons considered in this paper are much longer.

In this paper, we investigate several mathematical programming-based (utilitarian, leximin, approximate leximin) and ad hoc algorithms (greedy, round-robin) to allocate paths in conflicting graphs. We generalize our previous work [4] to the case of directed path allocation problems (DPAP) and consider another conflict expression that is based on resources. Note that a more detailed description of the work performed in [4] is presented later in the paper.

In another direction, there is a wide literature on Earth observation scheduling problems (EOSPs) [17]. In such problems, some observation requests have to be assigned to

satellites and scheduled for each satellite so that several constraints (e.g., temporal constraints related to the possible maneuvers of the satellites) are satisfied. Various criteria have been studied in the literature. Nevertheless, as the problem is generally over-constrained because of the number of requests to satisfy, a classical criterion is to optimize a (weighted) total reward provided by the satisfied requests. The Earth observation scheduling problem has a lot in common with the orbit slot allocation problem (OSAP) considered in this paper. Indeed, both problems involve observation requests posted by users, visibility windows, constraints related to the satellite disjunctive nature, etc. However, there are several differences between the two problems. First, in the orbit slot allocation problem, users want to "own" the satellite during some orbit portions in order to perform a set of observations. The targets to be observed during each slot are not precisely known in advance, which means that constraints about the satellites' maneuvers are irrelevant in the case of OSAP. Then, the requests' nature is different. In fact, in OSAP, the requests are composed of several slots possibly over several months ahead, and each slot is quite long as it is supposed to allow the user to perform several observations. In the case of EOSPs, there are many more requests but on a very short time horizon (a few days at most), and each request requires a very small amount of satellite time. Finally, fairness between users is essential in the case of OSAPs, whereas it is rarely considered in EOSPs. Two exceptions are the work described in [18], where the authors study a multi-objective EOSP and aim at maximizing the total profit and minimizing the maximum profit difference between each pair of users, and the work described in [19], where a heuristic method is proposed to solve the EOSP while taking into account fairness.

Using graphs in the context of EOSPs is not novel. In [20], an activity-on-node graph allows modeling of all the alternatives to satisfy observation requests by a set of satellites (one node is one opportunity to observe a request target by a satellite, and the edges allow conflicts between observation candidates to be represented). Then, maximizing the number of satisfied requests amounts to computing the maximum independent set of the graph.

## 3. Directed Path Allocation Problems

In this section, we define the so-called directed path allocation problem (DPAP), where agents' valuations of item bundles are represented as edge-weighted DAGs, as illustrated in Figure 1, and where the goal is to select one path in each DAG while satisfying set compatibility constraints over the selected paths. We first introduce some notation related to graphs and then formalize the generic problem we consider.

**Definition 1.** *A single-source single-sink edge-weighted DAG $g$ is a triple $\langle V_g, E_g, u_g \rangle$ such that:*

- *$V_g$ is a set of nodes; in our case, each node corresponds to an item that can be allocated to an agent, except for two specific nodes referred to as the source $s_g$ and the sink $t_g$;*
- *$E_g \subset V_g \times V_g$ is the set of arcs of the acyclic graph, with the assumption that $s_g$ and $t_g$ are, respectively, the unique source and sink of the graph; an arc $v_1 \rightarrow v_2$ indicates that items $v_1$ and $v_2$ can be selected sequentially;*
- *$u_g : E_g \rightarrow \mathbb{R}^+$ is a utility function that associates a weight to each arc of the graph to represent a preference over the combinations of item selections; we assume that $E_g$ contains an arc from $s_g$ to $t_g$ labeled by utility $0$, to deal with cases where no bundle of items can be selected in $g$.*

*In the following, the set of paths from $s_g$ to $t_g$ is denoted by $\Pi_g$.*

For each graph $g$ and each set of edges $X \subseteq E_g$, the utility of $X$ for $g$ is defined by $u_g(X) = \sum_{e \in X} u_g(e)$, which means that edge valuations are additive. As a result, each path from $s_g$ to $t_g$ in a graph $g$ is evaluated by summing the utilities of the traversed edges, and each DAG represents, in a compact manner, a set of valuations for bundles of items, as in combinatorial auctions.

**Definition 2.** *A Directed Path Allocation Problem (DPAP) is a tuple $\langle \mathcal{A}, \mathcal{G}, \mu, \phi \rangle$, where*

- *$\mathcal{A} = \{1, \ldots, n\}$ is a set of agents;*

- $\mathcal{G} = \{g_1, \ldots, g_m\}$ *is a set of single-source single-sink edge-weighted DAGs, as introduced in Definition* 1;
- $\mu : \mathcal{G} \to \mathcal{A}$ *maps each graph g in $\mathcal{G}$ to its owner a in $\mathcal{A}$; we also denote by $\mathcal{G}_a = \mu^{-1}(a)$ the set of graphs owned by agent a;*
- $\phi : \Pi_{g_1} \times \ldots \times \Pi_{g_m} \to \{0, 1\}$ *is a* path compatibility function *that indicates whether a combination of paths $(p_1, \ldots, p_m)$ (one path per graph) is feasible (value 1) or not (value 0).*

In a DPAP, the definition of the path compatibility function $\phi$ is related to the presence of items that cannot be shared by the agents. More precisely, a conflict between two paths represents the fact that assigning these paths to clients is infeasible (e.g., because some orbit slots overlap) or strongly undesirable for the constellation manager. A naive definition of the compatibility function is the list of combinations of paths that are compatible with each other. However, the number of paths in a DAG is exponential, which makes this definition impractical in the general case. Therefore, in the next sections, we propose and discuss different ways to define the compatibility function in a compact way.

**Example 1.** *Figure* 1 *illustrates a DPAP representing an orbit slot allocation problem. In such a problem, satellite orbit slots must be allocated to agents so that the latter can make several observations of a POI on Earth. In this example, we consider two agents* A *and* B *that each have one observation request, request* a *for agent* A *and request* b *for agent* B.

*Within the DPAP modeling framework, we consider a graph for each request: graph $g_a$ for request* a *and graph $g_b$ for request* b*. The nodes of these graphs are the orbit slot candidates for each request (slots $a_1$, $a_2$, and $a_3$ for request* a*, and slots $b_1$, $b_2$, $b_3$, and $b_4$ for request* b*). A path in a graph represents a way to satisfy the corresponding request. For instance, for satisfying request* a*, starting from $s_a$, one can either select first slot $a_1$ and then slot $a_2$, or select first slot $a_3$ and then slot $a_2$. Each edge has a utility that represents the reward for selecting slots in a given order. For instance, edges $s_a \to a_1$ and $s_a \to a_3$ have utilities equal to 0.2 and 0.5, respectively. This represents the fact that agent* A *prefers selecting slot $a_3$ rather than selecting slot $a_1$. Such a difference can be due to a satellite viewing angle that is better for $a_3$ than for $a_1$. Note that for a node, its incoming edges do not necessarily have the same utility value. For instance, the utility of edge $b_1 \to b_4$ is equal to 0.1, whereas the utility of edge $b_3 \to b_4$ is equal to 0.3.*

*The graph associated with request* a *contains three possible paths while the graph associated with request* b *contains five possible paths. We assume here that only* 10 *combinations of paths are allowed by the path compatibility function $\phi$ among the* 15 *possible ones. For instance, paths $\pi_{a,1}$ and $\pi_{b,2}$ are compatible ($\phi(\pi_{a,1}, \pi_{b,2}) = 1$) but paths $\pi_{a,1}$ and $\pi_{b,3}$ are not ($\phi(\pi_{a,1}, \pi_{b,3}) = 0$).*

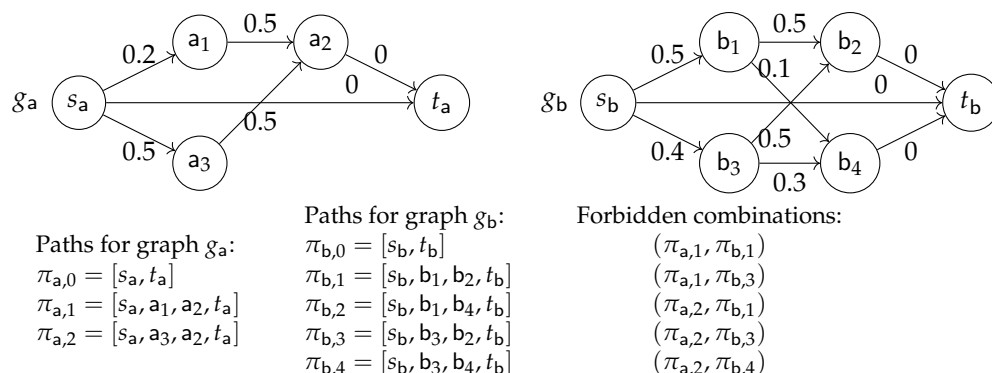

Paths for graph $g_a$:
$\pi_{a,0} = [s_a, t_a]$
$\pi_{a,1} = [s_a, a_1, a_2, t_a]$
$\pi_{a,2} = [s_a, a_3, a_2, t_a]$

Paths for graph $g_b$:
$\pi_{b,0} = [s_b, t_b]$
$\pi_{b,1} = [s_b, b_1, b_2, t_b]$
$\pi_{b,2} = [s_b, b_1, b_4, t_b]$
$\pi_{b,3} = [s_b, b_3, b_2, t_b]$
$\pi_{b,4} = [s_b, b_3, b_4, t_b]$

Forbidden combinations:
$(\pi_{a,1}, \pi_{b,1})$
$(\pi_{a,1}, \pi_{b,3})$
$(\pi_{a,2}, \pi_{b,1})$
$(\pi_{a,2}, \pi_{b,3})$
$(\pi_{a,2}, \pi_{b,4})$

**Figure 1.** Sample users' bundle valuations (or preferences) represented as a DPAP.

**Definition 3.** *For a DPAP $\langle \mathcal{A}, \mathcal{G}, \mu, \phi \rangle$, an* allocation *is a function $\pi$ that associates, with each graph $g \in \mathcal{G}$, one path $\pi(g)$ from $s_g$ to $t_g$ in $g$. If $\mathcal{G} = \{g_1, \ldots, g_m\}$, such an allocation is* valid *if and only if $\phi(\pi(g_1), \ldots, \pi(g_m)) = 1$ holds. Formally, $\pi(g)$ can be represented as a set of nodes in $V_g$. Indeed, as DAGs are manipulated, it is easy to reconstruct the edges successively traversed by the path from this set.*

**Definition 4.** *For a DPAP $\langle \mathcal{A}, \mathcal{G}, \mu, \phi \rangle$, the* global utility $u(\pi)$ *associated with an allocation $\pi$ is the sum of the utilities obtained in each graph, that is $u(\pi) = \sum_{g \in \mathcal{G}} u_g(\pi(g))$. The utility obtained for agent a is $u_a(\pi) = \sum_{g \in \mathcal{G}_a} u_g(\pi(g))$.*

**Definition 5.** *For a DPAP $\langle \mathcal{A}, \mathcal{G}, \mu, \phi \rangle$ involving n agents, the* leximin utility vector *associated with an allocation $\pi$ is the vector $\mathsf{lex}(\pi) = (\Lambda_1, \dots, \Lambda_n)$ that corresponds to vector $(u_1(\pi), \dots, u_n(\pi))$ sorted following an increasing order ($\Lambda_i \leq \Lambda_j$ holds for $i < j$).*

If $\pi$ and $\pi'$ denote two allocations for a given DPAP, and $\mathsf{lex}(\pi) = (\Lambda_1, \dots, \Lambda_n)$ and $\mathsf{lex}(\pi') = (\Lambda'_1, \dots, \Lambda'_n)$ are their associated leximin utility vectors, $\pi$ is strictly better than $\pi'$ with respect to the leximin criterion if there exists $k$ in $[1..n]$ such that $\Lambda_k > \Lambda'_k$ and for all $i < k$, $\Lambda_i = \Lambda'_i$. Note that leximin-based fair allocations allow the favoring of agents that are less satisfied.

The problems we consider in this paper are: (i) how to compute an optimal (utilitarian) valid allocation $\pi$ that maximizes $u(\pi)$, and (ii) how to compute an optimal fair valid allocation $\pi$ that maximizes $\mathsf{lex}(\pi)$.

**Example 2.** *In the graphs described in Example 1 and illustrated in Figure 1, the individual best paths for agents A and B are $\{s_a, a_3, a_2, t_a\}$ and $\{s_b, b_1, b_2, t_b\}$, respectively. They both have a utility equal to 1. However, these paths are not compatible according to the list of forbidden paths and cannot both belong to a valid allocation.*

*Figure 2a gives an example of a valid allocation $\pi_{\mathsf{ex}} = \{g_a \mapsto \{s_a, a_1, a_2, t_a\}, g_b \mapsto \{s_b, b_1, b_4, t_b\}\}$ for the DPAP introduced before. The global utility of $\pi_{\mathsf{ex}}$ is $u(\pi_{\mathsf{ex}}) = u(\pi_{\mathsf{ex}}(A)) + u(\pi_{\mathsf{ex}}(B)) = 0.7 + 0.6 = 1.3$. The leximin vector associated with $\pi_{\mathsf{ex}}$ is $\mathsf{lex}(\pi_{\mathsf{ex}}) = (0.6, 0.7)$: agent B has the lowest utility (0.6), and agent A's utility is equal to 0.7.*

*Figure 2b illustrates allocation $\pi_{\mathsf{util}} = \{g_a \mapsto \{s_a, a_3, a_2, t_a\}, g_b \mapsto \{s_b, b_1, b_4, t_b\}\}$ that maximizes the global utility: $u(\pi_{\mathsf{util}}) = 1.0 + 0.6 = 1.6$. The leximin vector is $\mathsf{lex}(\pi_{\mathsf{util}}) = (0.6, 1.0)$.*

*Figure 2c illustrates allocation $\pi_{\mathsf{lex}} = \{g_a \mapsto \{s_a, a_1, a_2, t_a\}, g_b \mapsto \{s_b, b_3, b_4, t_b\}\}$ that maximizes the leximin vector: $\mathsf{lex}(\pi_{\mathsf{lex}}) = (0.7, 0.7)$. The global utility associated with $\pi_{\mathsf{lex}}$ is lower: $u(\pi_{\mathsf{util}}) = 1.4$.*

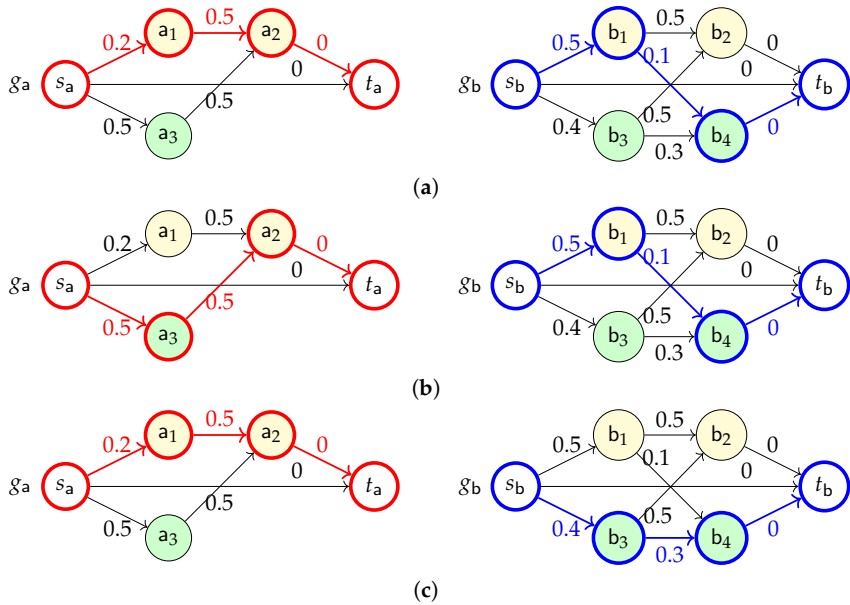

**Figure 2.** Examples of valid allocations for the DPAP described in Figure 1. (**a**) Illustration of allocation $\pi_{\mathsf{ex}}$ with the paths selected in graphs $g_a$ and $g_b$. (**b**) Allocation $\pi_{\mathsf{util}}$ that maximizes the global utility: $u(\pi_{\mathsf{util}}) = 1.6$. (**c**) Allocation $\pi_{\mathsf{lex}}$ that maximizes the leximin vector: $\mathsf{lex}(\pi_{\mathsf{lex}}) = (0.7, 0.7)$.

## 4. V-DPAP: Vertex-Constrained Directed Path Allocation Problems

In practice, the compatibility function $\phi$ that describes the allowed combinations of paths must be described in a compact way. We study the case where $\phi$ is simply defined by a set of conflicts between vertices, where each conflict corresponds to a subset of items that cannot all be simultaneously selected. For our target application related to booking orbit slots over a constellation of satellites, this is useful to model situations where two satellite slots required for two distinct booking requests are not compatible because they overlap and require the same satellite. The introduction of conflicts between vertices leads us to a specific case of DPAP called the vertex-constrained directed path allocation problem (V-DPAP). Note that V-DPAP is very close to the problem presented in [4].

### 4.1. Framework Definition

**Definition 6.** *A* Vertex-Constrained Directed Path Allocation Problem *(V-DPAP) is a DPAP* $\langle \mathcal{A}, \mathcal{G}, \mu, \phi \rangle$ *where function $\phi$ is defined by a set of conflicts $\mathcal{C}$ between vertices of the graph. Each conflict $\sigma \in \mathcal{C}$ is a non-empty set of vertices $V_\sigma$ that cannot all be selected by an allocation. Moreover, we assume that the vertices in $V_\sigma$ all belong to distinct graphs.*

*From this, function $\phi$ returns a value of $0$ for a selection of paths $(p_1, \ldots, p_m)$ if and only if there exists a conflict $\sigma \in \mathcal{C}$ such that all vertices in $V_\sigma$ are traversed by one path in $(p_1, \ldots, p_m)$. Formally, $\phi(p_1, \ldots, p_m) = 0$ if there exists $\sigma \in \mathcal{C}$ such that $V_\sigma \subseteq \bigcup_{i=1}^{m} V_{p_i}$, where $V_{p_i}$ denotes the set of vertices in path $p_i$.*

The previous definition covers both binary conflicts holding on two vertices and n-ary conflicts holding on any set of vertices. This differs from our initial framework, called PADAG, where only binary vertex conflicts were considered [4]. We will sometimes define a V-DPAP as a tuple $\langle \mathcal{A}, \mathcal{G}, \mu, \mathcal{C} \rangle$ equivalent to $\langle \mathcal{A}, \mathcal{G}, \mu, \phi \rangle$ since $\phi$ is non-ambiguously defined by the set of conflicts $\mathcal{C}$.

**Example 3.** *Figure 3 illustrates a V-DPAP that contains two conflicts, namely, conflict $\sigma_1 = \{a_2, b_2\}$ that invalidates any combination of paths traversing both $a_2$ and $b_2$, and conflict $\sigma_2 = \{a_3, b_3\}$ that invalidates any combination of paths traversing both $a_3$ and $b_3$. It can be shown that these conflicts lead to the same valid allocations as the ones provided in the DPAP of Figure 1.*

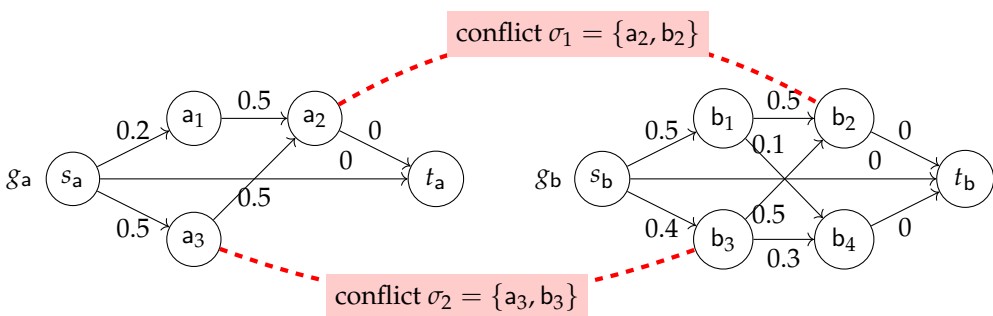

**Figure 3.** V-DPAP equivalent to the DPAP example of Figure 1; the set of vertex conflicts, represented as red hypernodes, gives a compact representation of the set of allowed combinations of paths.

### 4.2. Theoretical Complexity

**Proposition 1.** *For a V-DPAP, determining whether there exists a valid allocation $\pi$ such that utilitarian evaluation $u(\pi)$ is greater than or equal to a given value is NP-complete.*

**Proof.** First, the problem is NP since $u(\pi)$ is computable in polynomial time. Then, there exists a polynomial reduction of 3-SAT (which is NP-complete) to our problem. In a 3-SAT formula that contains $m$ clauses, each clause over the propositional variables $x, y, z$ can be represented as a weighted DAG $g$, where:

1. the set of nodes is $V_g = \{x, \neg x, y, \neg y, z, \neg z, s_g, t_g\}$,

2. the set of paths from $s_g$ to $t_g$ in $g$ corresponds to the set of truth values for $x, y, z$ that satisfy the clause (decision diagram representation),
3. the weight of every edge is set to 0, except for edges $s_g \to n$ where $n \neq t_g$, that have weight 1.

Last, for every propositional variable $x$, we can add one conflict $(n, n')$ for each pair of nodes labeled by the literals $x$ and $\neg x$ in two distinct graphs.

For instance, the 3-SAT problem $(x \vee y \vee z) \wedge (\neg x \vee y \vee \neg w)$ can be represented by the V-DPAP illustrated in Figure 4. Clause $(x \vee y \vee z)$ is translated into graph $g_1$ and clause $(\neg x \vee y \vee \neg w)$ into graph $g_2$. Vertices linked by dashed edges correspond to conflicts.

Then, as one path is selected in each graph and as there are $m$ graphs, determining whether there exists a valid allocation $\pi$ such that $u(\pi) \geq m$, with $m$ the number of clauses in the 3-SAT formula, is equivalent to finding a solution that satisfies all the clauses, hence the NP-completeness result given that all operations used in the transformation are polynomial. $\square$

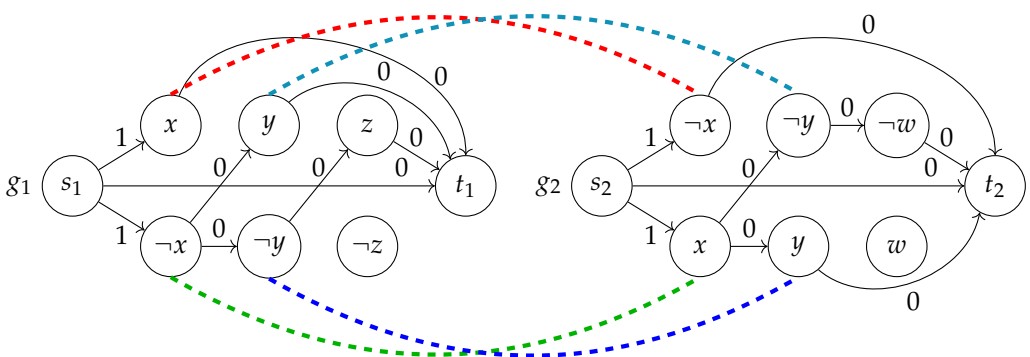

**Figure 4.** V-DPAP associated with the 3-SAT instance $(x \vee y \vee z) \wedge (\neg x \vee y \vee \neg w)$. Nodes in conflict are linked through a dashed edge.

**Proposition 2.** *For a V-DPAP, it is NP-complete to decide whether there exists a valid allocation whose leximin evaluation is greater than or equal to a given utility vector. The proposition holds even if there is a unique graph per agent.*

**Proof.** In the general case, it suffices to consider a problem involving a unique agent owning all the graphs, and to use the result of the previous proposition. If there is a unique graph per agent, it suffices to use the exact same 3-SAT encoding as before. Then, it is possible to show that there exists a valid allocation whose leximin evaluation is greater than or equal to $(1, 1, \ldots, 1)$ if and only if there exists a solution for the 3-SAT problem. Furthermore, the leximin evaluation of an allocation $\pi$ can be computed in polynomial time, hence the NP-completeness result. $\square$

## 5. R-DPAP: Resource-Constrained Directed Path Allocation Problems

The V-DPAP framework allows the posting of constraints on the simultaneous selection of items from different graphs. This is particularly relevant when the items correspond to tasks that require disjunctive resources over a given time frame. In this case, if two tasks $i$ and $j$ need to book the same resource over two time intervals $[ws(i), we(i)]$ and $[ws(j), we(j)]$, respectively, and if these two time intervals overlap, then a conflict $\{i, j\}$ can be defined. However, in practice, tasks $i$ and $j$ can be temporally flexible and can require the resource only during limited durations $d(i)$ and $d(j)$, respectively. In this case, even if time windows $[ws(i), we(i)]$ and $[ws(j), we(j)]$ overlap, tasks $i$ and $j$ may still be compatible. Such specifications are useful for our target application, where an agent may request a satellite only during 2 or 3 min over the whole 10 min pass of that satellite over the area of interest. This section introduces another extension of DPAP that is adapted to path allocation for items corresponding to such temporally flexible tasks.

*5.1. Framework Definition*

**Definition 7.** *A Resource-Constrained Directed Path Allocation Problem (R-DPAP) is a DPAP $\langle \mathcal{A}, \mathcal{G}, \mu, \phi \rangle$ where function $\phi$ is defined by:*

- *a set of disjunctive resources $\mathcal{R} = \{r_1, \ldots, r_p\}$;*
- *for each graph $g \in \mathcal{G}$, a triple $(ws_g, we_g, c_g)$ such that:*
  - *$ws_g : V_g \to \mathbb{N}$ and $we_g : V_g \to \mathbb{N}$ associate a start date and an end date, respectively, that together define a time window for each item;*
  - *$c_g : V_g \to \mathcal{R} \cup \{r_\varnothing\}$ returns the resource required for each item. For any vertex $v \in V_g$, $c_g(v) = r_\varnothing$ indicates that $v$ does not require any resource in $\mathcal{R}$. In particular, the source and sink nodes do not consume any resource. Moreover, we assume that for two items $v$ and $v'$ belonging to the same graph and requiring the same resource in $\mathcal{R}$, the time windows of $v$ and $v'$ do not overlap;*
  - *$d_g : V_g \to \mathbb{N}$ associates a duration with each item; resource $c_g(v)$ must be used during $d_g(v)$ time units within time window $[ws_g(v), we_g(v)]$ without any interruption (non-preemptive consumption).*

*From this, function $\phi$ returns a value of $1$ for a path allocation if and only if, given the items selected by the paths, there exists a way to schedule the consumptions over the disjunctive resources in $\mathcal{R}$ (see Definition 8).*

**Definition 8.** *In an R-DPAP $\langle \mathcal{A}, \mathcal{G}, \mu, \phi \rangle$, an allocation $\pi$ is valid if and only if for each graph $g \in \mathcal{G}$, there exists a function $\tau_{\pi,g} : \pi(g) \to \mathbb{N}$ that assigns a start date to each node $v$ in $\pi(g)$ such that:*

- *for all graphs $g \in \mathcal{G}$, for all nodes $v \in \pi(g)$, $\tau_{\pi,g}(v) \geq ws_g(v)$ and $\tau_{\pi,g}(v) + d_g(v) \leq we_g(v)$;*
- *there is no conflict for nodes in $\pi(g)$ with respect to resource consumption. Formally, for each pair of distinct graphs $g$ and $g'$, for each node $v \in \pi(g)$ and each node $v' \in \pi(g')$ such that $c_g(v) = c_{g'}(v')$ and $c_g(v) \neq r_\varnothing$ (i.e., $v$ and $v'$ consume the same resource in $\mathcal{R}$), either $\tau_{\pi,g}(v) + d_g(v) \leq \tau_{\pi,g'}(v')$ or $\tau_{\pi,g'}(v') + d_{g'}(v') \leq \tau_{\pi,g}(v)$ holds.*

**Example 4.** *We reuse the orbit slot allocation problem whose graph is given in Figure 1, and where two requests a and b are involved. We assume here that each request requires two observation slots of duration $2$. For both requests, the first slot must occur around time $3$ and the second slot around time $9$. We consider two satellites $sat_1$ and $sat_2$. For request a, there are two time windows around time $3$ during which satellites pass over the target area of a: time window $a_1 = [1, 4]$ for satellite $sat_1$ and time window $a_3 = [2, 4]$ for satellite $sat_2$. Around time $9$, only satellite $sat_1$ passes over the target area, which results in time window $a_2 = [7, 10]$. Similarly, for request b, time windows $b_1 = [2, 5]$ and $b_3 = [1, 4]$ allow the target area to be observed around time $3$ with satellites $sat_1$ and $sat_2$, respectively. Time windows $b_2 = [8, 10]$ and $b_4 = [9, 12]$ are available for observing around time $9$. Such a problem can be represented through the R-DPAP illustrated in Figure 5. Each satellite can be seen as a resource. Each request is represented through a graph: graph $g_a$ for request a and graph $g_b$ for request b. The nodes in the graph correspond to the time windows associated with each request and each satellite. For instance, node $a_1$ in graph $g_a$ corresponds to time window $a_1 = [1, 4]$. Formally, $d_{g_a}(a_1) = 2$ (because an observation duration equal to $2$ is required), $ws_{g_a}(a_1) = 1$, $we_{g_a}(a_1) = 4$ (corresponding to time window $[1, 4]$) and $c_{g_a}(a_1) = sat_1$.*

**Example 5.** *Figure 6a illustrates an allocation for the problem presented in Example 4. A valid allocation could be $\pi_{ex} = \{g_a \mapsto \{s_a, a_1, a_2, t_a\}, g_b \mapsto \{s_b, b_1, b_4, t_b\}\}$. In fact, we can consider two functions $\tau_{\pi_{ex}, g_a}$ and $\tau_{\pi_{ex}, g_b}$ that assign start dates to nodes in $\pi_{ex}$ without any conflict in resources. As illustrated in Figure 6b, it is possible to have $\tau_{\pi_{ex}, g_a}(a_1) = 1$ (i.e., a slot starting at time $1$ and ending at time $3$ is booked within time window $a_1$), $\tau_{\pi_{ex}, g_a}(a_2) = 7$, $\tau_{\pi_{ex}, g_b}(b_1) = 3$, and $\tau_{\pi_{ex}, g_b}(b_4) = 9$, which results in a non-conflicting access to the resources $sat_1$ and $sat_2$.*

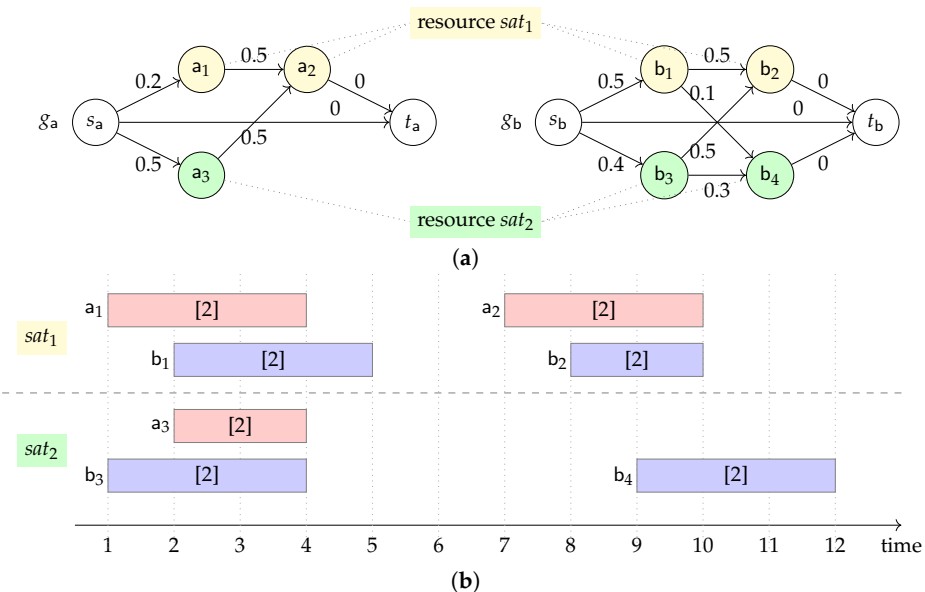

**Figure 5.** Orbit slot allocation problem involving two satellites *sat*$_1$ and *sat*$_2$ and two agents A and B that each have one request, denoted a and b, respectively. Two observation slots with a duration equal to 2 must be allocated for each request (represented by *[2]* in each observation slot). The first orbit slot of each request should be around time 3 and the second one around time 9. (**a**) Graphs *g*$_a$ and *g*$_b$ representing the requests and resources of Example 4. (**b**) Description of the resources, time windows, and durations associated with the vertices of graphs *g*$_a$ and *g*$_b$.

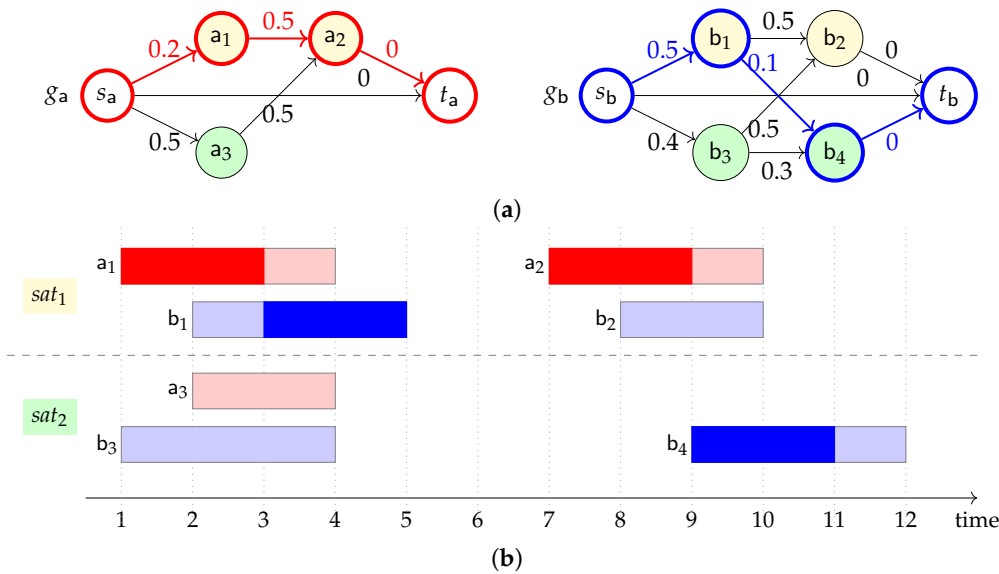

**Figure 6.** Valid allocation example, $\pi_{ex}$, for the R-DPAP described in Example 4. (**a**) Illustration of allocation $\pi_{ex}$ with the paths selected for graphs *g*$_a$ and *g*$_b$. (**b**) Start dates that allow the selection of the nodes of $\pi_{ex}$ without any conflict in resources.

### 5.2. Theoretical Complexity

**Proposition 3.** *The* R-DPAP-UTIL-DEC *problem, which consists in determining whether, for a given R-DPAP problem, there exists an allocation $\pi$ and start time functions $\tau_{\pi,g}$ such that $\pi$ is valid and utilitarian evaluation $u(\pi)$ is greater than or equal to a given value, is NP-complete.*

**Proof.** Given an R-DPAP, an allocation $\pi$ for it, a start time function $\tau_{\pi,g}$ for each graph $g$, and a utility lower bound $L$, verifying that the scheduling constraints are satisfied and that

$u(\pi)$ is greater than or equal to $L$ is polynomial. This proves that R-DPAP-UTIL-DEC is in class NP.

To prove the NP-completeness of R-DPAP-UTIL-DEC, we rely on the fact that the one-machine scheduling problem with release dates and due dates in which the objective is to minimize the maximum lateness of jobs is NP-complete [21].

Let $\langle A, P, R, D \rangle$ be such a problem where:

- $A = \{a_1, \ldots, a_n\}$ is a set of activities;
- $P : A \to \mathbb{N}$ is a function that assigns a processing time to each activity of $A$;
- $R : A \to \mathbb{N}$ is a function that assigns a release date to each activity of $A$;
- $D : A \to \mathbb{N}$ is a function that assigns a due date to each activity of $A$.

The objective of the problem is to define a function $\sigma : A \to \mathbb{N}$ that assigns a start date $\sigma(a)$ to each activity $a$ in $A$ such that:

- the release dates are satisfied, i.e., $\forall a \in A, \sigma(a) \geq R(a)$;
- the machine performs at most one activity at each time step, i.e., $\forall a_i, a_j \in A^2$ with $i \neq j$, either $\sigma(a_i) + R(a_i) \leq \sigma(a_j)$ or $\sigma(a_j) + R(a_j) \leq \sigma(a_i)$ holds;
- the maximum lateness $L_{max}$ is minimized, where $L_{max} = \max_{i=1}^{n}(\sigma(a_i) + P(a_i) - D(a_i))$.

In the associated decision problem, we consider a bound $l$, and the objective is to decide if it is possible to define $\sigma$ such that $L_{max} \leq l$.

Such a problem can be transformed to an R-DPAP as follows:

- we consider a unique resource $r$;
- we consider an agent $agent_a$ for each activity $a$ in $A$;
- for each activity $a$ in $A$, we consider the graph $g_a$ (illustrated in Figure 7a) that belongs to agent $agent_a$ and that has the following features:
  - its set of vertices is composed of three nodes: $s_a$, $t_a$, and $v_a$;
  - its set of edges is composed of $(s_a, v_a)$ with a utility equal to 1, and $(s_a, t_a)$, $(v_a, t_a)$ that both have a null utility;
  - as illustrated in Figure 7b, node $v_a$ requires resource $r$ during $D(a)$ time units within time window $[R(a), D(a) + l]$;
- the obtained R-DPAP is $\langle \{agent_a \mid a \in A\}, \{r\}, \{g_a \mid a \in A\}, \mu \rangle$, with $\mu$ a function that assigns, for each activity $a$ in A, agent $agent_a$ to graph $g_a$.

The maximum lateness is lower than or equal to $l$ in the machine scheduling problem if and only if there exists a valid allocation $\pi$ with a utility greater than or equal to $n$. Indeed, to reach such a utility value, the paths selected in the $n$ graphs must each have a utility equal to 1. The selection of such paths indicates that all activities $v_a$ can be scheduled on the unique resource while satisfying the release date and the due date, to which is added the lateness bound $l$. □

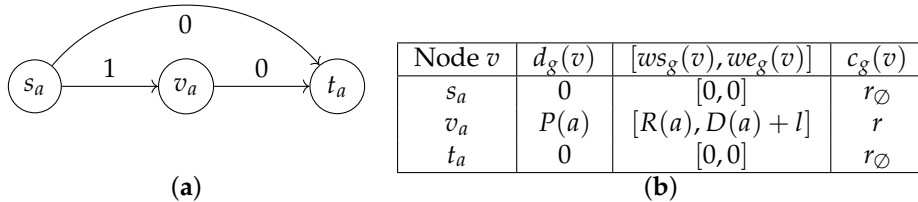

| Node $v$ | $d_g(v)$ | $[ws_g(v), we_g(v)]$ | $c_g(v)$ |
|---|---|---|---|
| $s_a$ | 0 | $[0,0]$ | $r_\varnothing$ |
| $v_a$ | $P(a)$ | $[R(a), D(a) + l]$ | $r$ |
| $t_a$ | 0 | $[0,0]$ | $r_\varnothing$ |

(**a**)                                                         (**b**)

**Figure 7.** R-DPAP part generated for each activity $a$ in $A$. (**a**) Graph generated for each activity $a$ in $A$. (**b**) Description of nodes in the graph generated for each activity $a$ in $A$.

**Proposition 4.** *The* R-DPAP-LEX-DEC *problem, which consists in determining whether, for a given R-DPAP problem, there exists an allocation $\pi$ and start time functions $\tau_{\pi,g}$ such that $\pi$ is valid and its leximin evaluation is greater than or equal to a given utility vector, is NP-complete.*

**Proof.** By using the same encoding as in the previous proof, there exists a solution such that $L_{\max} \leq l$ if and only if the leximin-optimal allocation has a value greater than or equal to $(1, 1, \ldots, 1)$. Further, the leximin evaluation of an allocation $\pi$ can be computed in polynomial time, hence the NP-completeness result. □

*5.3. Relationship between R-DPAP and V-DPAP*

An R-DPAP combines a path selection problem and a scheduling problem over the resources used by the selected items. In the following, we show that it is possible to transform an R-DPAP into an equivalent V-DPAP by generating a set of item selection conflicts that is equivalent to the set of selections forbidden by the scheduling problem.

To illustrate this point, let us consider the example given in Figure 8, that involves four requests: $a, b, c, d$. It is first possible to decompose the scheduling problem of the R-DPAP into a set of subproblems containing items that may be in competition for using a given resource (gray rectangles depicted in the figure). For example, items $a_4$ and $b_5$ belong to the same subproblem because their time windows overlap, and items $b_5$ and $c_1$, whose time windows do not overlap, also belong to the same subproblem because the presence of items $a_4$ and $d_1$ creates an indirect interaction between $b_5$ and $c_1$. More formally, to compute the content of these scheduling subproblems, we can build, for each resource $r$, the graph $G_r$ containing one node per item and one edge between item $i$ and item $j$ if and only if the time windows of $i$ and $j$ overlap. Then, the scheduling subproblems to consider correspond to the connected components of graph $G_r$. In Figure 8, we obtain three connected components for resource $sat_1$, namely, $\{a_1, b_1\}$, $\{a_2, b_2\}$, and $\{a_4, b_5, c_1, d_1\}$, and three connected components for resource $sat_2$, namely, $\{a_3, b_3\}$, $\{b_4\}$, and $\{a_5, b_6, c_2, d_2\}$.

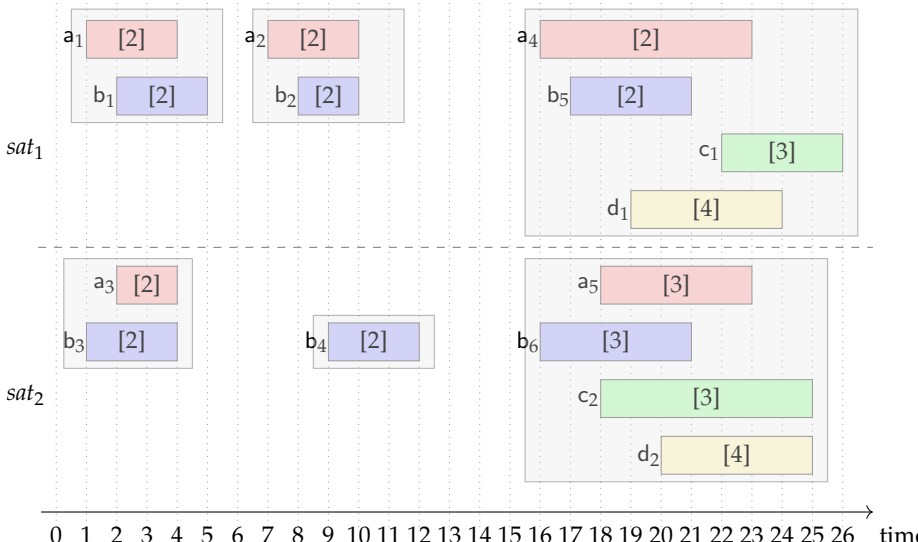

**Figure 8.** Orbit slot allocation problem involving two satellites $sat_1$ and $sat_2$ and four requests a, b, c, d posted by four agents A, B, C, D; the duration associated with each item is also indicated (e.g., a duration of 2 time units for item $a_1$ and a duration of 3 time units for item $c_1$).

After these steps, for each component $\Gamma$ obtained, we can compute the set of *minimal scheduling conflicts* associated with $\Gamma$. This set contains all the sets $S \subseteq \Gamma$ such that, (1) there is no feasible schedule performing all the tasks in $S$ while respecting their time window and duration constraints, and (2) set $S$ is minimal for inclusion, that is, for every set $S' \subset S$, there exists a way to schedule all the tasks in $S'$. To compute these minimal conflicts, we proceed as follows.

- We consider the non-empty subsets $S$ of $\Gamma$ one by one, following an increasing cardinality order. For a given set $S$, if there exists a subset $S' \subset S$ of size $|S| - 1$ such that $S'$ is a conflict, $S$ is marked as being a conflict but is not added to the set of minimal

conflicts. Otherwise, we test whether there exists a schedule containing all the tasks in $S$. If not, $S$ is marked as a conflict and added to the set of minimal conflicts.
- To determine whether there exists a schedule containing all the items in a set $S$, we use a dynamic programming algorithm. More precisely, we consider the subsets $S'$ of $S$ following an increasing cardinality order and we determine, for each subset $S' \subseteq S$, the minimum time $mt(S')$ at which all items in $S'$ can be served in a feasible schedule. To do this, we start from $mt(\varnothing) = -\infty$ and apply recursive formulas. If item $i \in S'$ belongs to graph $g$ and is the last item visited, the minimum time at which the visit of $i$ can end is given by $mt(S', i) = \max(mt(S' \setminus \{i\}), ws_g(i)) + d_g(i)$, and visiting $i$ at the latest position among the items in $S'$ is feasible if and only if $mt(S', i) \leq we_g(i)$. From this, the minimum time $mt(S')$ at which all items in $S'$ can be served in a feasible schedule is given by $mt(S') = \min_{i \in S' \mid mt(S',i) \leq we_g(i)} mt(S', i)$. It can be shown that at the end of the process, all the items in $S$ can be scheduled if and only if $mt(S) < +\infty$. The dynamic programming algorithm described before has a time complexity that is exponential in the size of $S$; however, the number of requests is low for the practical application we are targeting.

**Example 6.** *For the example given in Figure 8, the set of minimal conflicts obtained is*

$$\{\{a_2, b_2\}, \{a_4, b_5, c_1, d_1\}, \{a_3, b_3\}, \{a_5, b_6, d_2\}, \{b_6, c_2, d_2\}\}$$

*Such conflicts are equivalent to the constraints of the initial scheduling problem.*

The method described before allows us to transform an R-DPAP $\mathcal{P}$ into a V-DPAP $\mathcal{P}'$ that contains the exact same set of items as $\mathcal{P}$ and has the same graph topology as $\mathcal{P}$, and where the conflicts in $\mathcal{P}'$ are those obtained by preprocessing the scheduling problem of $\mathcal{P}$. In the following, given the (restricted) number of requests in our target application, we consider that such a transformation from R-DPAP to V-DPAP can be used and we focus on the definition of algorithms for solving V-DPAP.

## 6. V-DPAP Solution Methods

We propose here several allocation schemes for V-DPAP. Some of them are based on integer linear programming (ILP) and mixed integer linear programming (MILP), so we first introduce decision variables and constraints for these models. For any DAG $g = \langle V_g, E_g, u_g \rangle$, we define binary variables $x_e \in \{0,1\}$, for any $e \in E_g$, stating whether edge $e$ is selected in the path defining the solution bundle. We also use auxiliary binary variables $\beta_v$, stating whether node $v$ is selected in solution path $\pi(g)$, i.e., $\beta_v = 1$ if $v \in \pi(g)$, and 0 otherwise. For any node $v$ in $V_g$, we denote by $\mathsf{In}(v)$ (respectively $\mathsf{Out}(v)$) its set of incoming (respectively outcoming) edges. In all ILP models introduced hereafter, we impose constraints (1)–(3) to define all the possible paths, (4) and (5) to account for item selection conflicts, (6) to ensure that sources and sinks are selected, and (7) to define the edge selection variables.

$$\sum_{e \in \mathsf{In}(v)} x_e = \sum_{e \in \mathsf{Out}(v)} x_e, \quad \forall g \in \mathcal{G}, \forall v \in V_g \setminus \{s_g, t_g\} \tag{1}$$

$$\sum_{e \in \mathsf{Out}(s_g)} x_e = 1, \quad \forall g \in \mathcal{G} \tag{2}$$

$$\sum_{e \in \mathsf{In}(t_g)} x_e = 1, \quad \forall g \in \mathcal{G} \tag{3}$$

$$\sum_{e \in \mathsf{In}(v)} x_e = \beta_v, \quad \forall g \in \mathcal{G}, \forall v \in V_g \setminus \{s_g, t_g\} \tag{4}$$

$$\sum_{v \in \sigma} \beta_v \leq |\sigma| - 1, \quad \forall \sigma \in \mathcal{C} \tag{5}$$

$$\beta_{s_g} = \beta_{t_g} = 1, \quad \forall g \in \mathcal{G} \tag{6}$$

$$x_e \in \{0,1\}, \quad \forall a \in \mathcal{A}, \forall g \in \mathcal{G}_a, \forall e \in E_g \tag{7}$$

### 6.1. Utilitarian Allocation (util)

The classical approach to allocation is the utilitarian one. This consists in finding the allocation that maximizes the sum of utilities of all selected paths. This corresponds to solving the integer linear program $P_{\text{util}}(\langle \mathcal{A}, \mathcal{G}, \mu, \mathcal{C} \rangle)$ composed of constraints (1)–(7) and the objective function given below:

$$\textbf{maximize} \quad \sum_{a \in \mathcal{A}} \sum_{g \in \mathcal{G}_a} \sum_{e \in E_g} u_g(e) \cdot x_e \tag{8}$$

The resulting allocation $\pi$ is decoded from the $\beta_v$ variables. Formally, for all $g \in \mathcal{G}$, $\pi(g) = \{v \in V_g \mid \beta_v = 1\}$.

**Example 7.** *In Figure 3, the utilitarian allocation is $\pi_{\text{util}} = \{a \mapsto \{s_a, a_3, a_2, t_a\}$, $b \mapsto \{s_b, b_1, b_4, t_b\}\}$, with utility $u(\pi_{\text{util}}) = u_a(\pi_{\text{util}}) + u_b(\pi_{\text{util}}) = 1.0 + 0.6 = 1.6$.*

### 6.2. Leximin Allocation (lex)

Beyond utilitarianism, one way to implement fair allocation and Pareto-optimality is to consider the *leximin* rule, that selects, among all possible allocations, an allocation leading to the best utility profiles with respect to the leximin order [22]. More precisely, let $z = [z_1, \ldots, z_n]$ be the utility vector, where each component $z_a \in [0, Z_a]$ represents the utility for agent $a \in \mathcal{A}$. $Z_a$ denotes here the best utility value for user $a$ considered alone, i.e., for the mono-agent problem, where the best path can be chosen for each graph $g \in \mathcal{G}_a$. In leximin optimization, the objective is to lexicographically maximize vector $\Lambda = [\Lambda_1, \ldots, \Lambda_n]$ obtained after ordering $[z_1, \ldots, z_n]$ following an increasing order. Such a leximin rule can be implemented through a sequence of ILP [23]. We adapt here such a procedure to the specific case of V-DPAP. Suppose we have already optimized over the first $K - 1$ components $[\Lambda_1, \ldots, \Lambda_{K-1}]$ of $\Lambda$, for $K \in [1..n]$. Then, one can use the MILP presented thereafter to optimize the $K^{\text{th}}$ component $\Lambda_K$ of the leximin profile.

In this MILP model, variable $\lambda$ represents the utility optimized at level $K$ in $\Lambda$, with $\lambda \in [\Lambda_{K-1}, \max_{a \in \mathcal{A}} Z_a]$, using convention $\Lambda_0 = 0$. Variable $y_{ak}$ is a binary variable that takes value 1 if agent $a \in \mathcal{A}$ plays the role of the agent associated with level $k \in [1..K-1]$ in $[\Lambda_1, \ldots, \Lambda_{K-1}]$, and 0 otherwise. Constraint (10) computes the utility associated with each agent. Constraints (11) and (12) ensure that a unique agent is associated with each level $k \in [1..K-1]$ already dealt with. Constraint (13) ensures that the utility obtained for the agent associated with level $k \in [1..K-1]$ must not be less than $\Lambda_k$. Last, together with the objective function, Constraint (14) ensures that $\lambda$ will be equal to the minimum utility value obtained for the agents that are not associated with levels $[1..K-1]$ in $\Lambda$. In this constraint, $M = \max_{a \in \mathcal{A}} Z_a$ is used to ignore the agents associated with levels strictly lower than $K$ when optimizing $\lambda$ (big-M formulation). In the end, the optimization of $\Lambda_K$ can be performed using program $P_{\text{lex}}(\langle \mathcal{A}, \mathcal{G}, \mu, \mathcal{C} \rangle, K, [\Lambda_1, \ldots, \Lambda_{K-1}])$ that is composed of constraints (1)–(7) and the additional constraints and objective function given below:

$$\textbf{maximize} \quad \lambda \tag{9}$$

$$z_a = \sum_{g \in \mathcal{G}_a} \sum_{e \in E_g} u_g(e) \cdot x_e, \quad \forall a \in \mathcal{A} \tag{10}$$

$$\sum_{a \in \mathcal{A}} y_{ak} = 1, \quad \forall k \in [1..K-1] \tag{11}$$

$$\sum_{k \in [1..K-1]} y_{ak} \leq 1, \quad \forall a \in \mathcal{A} \tag{12}$$

$$z_a \geq \sum_{k \in [1..K-1]} \Lambda_k \cdot y_{ak}, \quad \forall a \in \mathcal{A} \tag{13}$$

$$\lambda \leq z_a + M \sum_{k \in [1..K-1]} y_{ak}, \quad \forall a \in \mathcal{A} \tag{14}$$

$$z_a \in [0, Z_a], \ \forall a \in \mathcal{A} \tag{15}$$

$$y_{ak} \in \{0, 1\}, \ \forall a \in \mathcal{A}, \forall k \in [1..K-1] \tag{16}$$

$$\lambda \in [\Lambda_{K-1}, \max_{a \in \mathcal{A}} Z_a] \tag{17}$$

To implement the leximin rule, it then suffices to solve a sequence of $P_{\text{lex}}$ problems for $K \in \mathcal{A}$ to optimize the value of each component of the utility profile, as presented in Algorithm 1.

---

**Algorithm 1:** Leximin algorithm.

---

**Data:** A V-DPAP $\langle \mathcal{A}, \mathcal{G}, \mu, \mathcal{C} \rangle$
**Result:** A leximin-optimal path allocation $\pi$
1 **for** $K = 1$ *to* $|\mathcal{A}|$ **do**
2    $(\lambda^*, sol) \leftarrow$ solve $P_{\text{lex}}(\langle \mathcal{A}, \mathcal{G}, \mu, \mathcal{C} \rangle, K, [\Lambda_1, \ldots, \Lambda_{K-1}])$
3    $\Lambda_K \leftarrow \lambda^*$
4 **for** $g \in \mathcal{G}$ **do** $\pi(g) \leftarrow \{v \in V_g \mid sol(\beta_v) = 1\}$
5 **return** $\pi$

---

**Example 8.** *For the example in Figure 3, the leximin-optimal allocation is* $\pi_{\text{lex}} = \{g_{\mathsf{a}} \mapsto \{s_{\mathsf{a}}, a_1, a_2, t_{\mathsf{a}}\}, g_{\mathsf{b}} \mapsto \{s_{\mathsf{b}}, b_3, b_4, t_{\mathsf{b}}\}\}$, *with utility vector* $(u_{\mathsf{A}}(\pi_{\text{lex}}), u_{\mathsf{B}}(\pi_{\text{lex}})) = (0.7, 0.7)$.

*6.3. Approximate Leximin Allocation (*a-lex*)*

The previous model implements an exact leximin rule, and thus enforces fairness in the resulting allocation. However, it may not scale well when increasing the number of agents and edges. This is why we provide an approximate version of the computation of the leximin based on an iterated maximin scheme. This approach considers at each step a minimum utility $\Delta_a \geq 0$ for some agents and maximizes the worst utility among the remaining agents, for which we arbitrarily assume $\Delta_a = -1$. The problem to solve, referred to as $P_{\text{a-lex}}(\langle \mathcal{A}, \mathcal{G}, \mu, \mathcal{C} \rangle, \Delta)$, is the following one:

$$\textbf{maximize} \quad \delta \tag{18}$$

$$\text{such that} \quad (1), (2), (3), (4), (5), (6), (7)$$

$$\delta \leq \sum_{g \in \mathcal{G}_a} \sum_{e \in E_g} u_g(e) x_e, \quad \forall a \in \mathcal{A} \mid \Delta_a = -1 \tag{19}$$

$$\sum_{g \in \mathcal{G}_a} \sum_{e \in E_g} u_g(e) x_e \geq \Delta_a, \quad \forall a \in \mathcal{A} \mid \Delta_a \neq -1 \tag{20}$$

$$\delta \in \mathbb{R}^+ \tag{21}$$

The solution method then consists in optimizing in an iterative manner, as for leximin. As sketched in Algorithm 2, at each iteration (one per agent), $P_{\text{a-lex}}$ is solved, one worst agent $\hat{a}$ is determined, and its minimum utility $\Delta_{\hat{a}}$ is fixed. The main difference with $P_{\text{lex}}$, is that at each iteration in $P_{\text{a-lex}}$ the position of an agent in the order is implicitly determined once for the whole algorithm, while in $P_{\text{lex}}$ the order can be revised at each iteration. Moreover, if any equality occurs at line 5 to determine the worst agent (case $|S| > 1$), one may rely on some heuristic or arbitrary choice. Thus, $P_{\text{a-lex}}$ is an approximation of $P_{\text{lex}}$ that contains fewer variables and constraints.

**Example 9.** *The approximate leximin allocation for the example in Figure 1 is* $\pi_{\text{a-lex}} = \{g_{\mathsf{a}} \mapsto \{s_{\mathsf{a}}, a_1, a_2, t_{\mathsf{a}}\}, g_{\mathsf{b}} \mapsto \{s_{\mathsf{b}}, b_3, b_4, t_{\mathsf{b}}\}\}$, *with utility vector* $(u_{\mathsf{A}}(\pi_{\text{a-lex}}), u_{\mathsf{B}}(\pi_{\text{a-lex}})) = (0.7, 0.7)$. *This is the same as* $\pi_{\text{lex}}$, *but in the general case,* $\pi_{\text{a-lex}}$ *and* $\pi_{\text{lex}}$ *can differ.*

---

**Algorithm 2:** Approximate leximin algorithm.

---

**Data:** A V-DPAP $\langle \mathcal{A}, \mathcal{G}, \mu, \mathcal{C} \rangle$
**Result:** An iterated maximin-optimal allocation $\pi$

1  $\Delta \leftarrow [-1, \ldots, -1]$
2  **for** $K = 1$ *to* $|\mathcal{A}|$ **do**
3  $\quad (\delta^*, sol) \leftarrow$ solve $P_{\text{a-lex}}(\langle \mathcal{A}, \mathcal{G}, \mu, \mathcal{C} \rangle, \Delta)$
4  $\quad S \leftarrow \underset{a \in \mathcal{A} \mid \Delta_a = -1}{\operatorname{\textbf{argmin}}} \sum_{g \in \mathcal{G}_a} \sum_{e \in E_g} u_g(e) sol(x_e)$
5  $\quad \hat{a} \leftarrow$ choose an agent $a$ in $S$
6  $\quad \Delta_{\hat{a}} \leftarrow \delta^*$
7  **for** $g \in \mathcal{G}$ **do** $\pi(g) \leftarrow \{v \in V_g \mid sol(\beta_v) = 1\}$
8  **return** $\pi$

---

### 6.4. Greedy Allocation (greedy)

For very fast decisions, approximate leximin might still be too slow. In such cases, a greedy approach can quickly provide valid allocations. The main idea of greedy path allocation is to iterate over the set of graphs. At each step, one graph $g^*$ that has the best utility path is selected and this path is chosen as $\pi(g^*)$. Moreover, given the nodes already selected and the new ones in $\pi(g^*)$, all the nodes in the other graphs that are in conflict are deactivated. Graph $g^*$ is then removed, and the process continues until there is no more graphs to consider. This process ensures that constraints (1)–(6) are met. Determining the best path in a DAG $g$ has a linear time complexity $\mathcal{O}(|E_g| + |V_g|)$ [24]. Obviously, greedy is equivalent to utilitarian when there is no conflict between graphs. Indeed, greedy will return the best path for each graph, which is the best utilitarian solution in such settings. Moreover, if there are no ties when selecting the best path for each graph, then this greedy approach leads to a Nash equilibrium, where no agent can improve its utility without a negative impact on other agents. This is equivalent to the *Nashify* procedure from [13] in the context of congestion games, with only one turn. We will see in the experiments that this equilibrium is far from being fair.

**Example 10.** *For the example in Figure 3, there is a path of value* 1 *in the two graphs $g_{\mathsf{a}}$ and $g_{\mathsf{b}}$. If the best path in $g_{\mathsf{a}}$ is chosen first, then the allocation obtained in the end is $\pi_{\text{greedy}} = \{g_{\mathsf{a}} \mapsto \{s_{\mathsf{a}}, \mathsf{a}_3, \mathsf{a}_2, t_{\mathsf{a}}\}, g_{\mathsf{b}} \mapsto \{s_{\mathsf{b}}, \mathsf{b}_1, \mathsf{b}_4, t_{\mathsf{b}}\}\}$, with global utility $u(\pi_{\text{greedy}}) = u_{\mathsf{A}}(\pi_{\text{greedy}}) + u_{\mathsf{B}}(\pi_{\text{greedy}}) = 1.0 + 0.6 = 1.6$ and utility vector $(1.0, 0.6)$.*

### 6.5. Round-Robin Allocations (p-rr and n-rr)

One fast approach to the fair allocation of indivisible goods is *round-robin*. This consists in making each agent choose in turn (in a predefined fixed order) one item (depending on the preferences) until there is no more item to allocate. It is polynomial in the number of agents and items. In our case, one may consider two kinds of items to allocate: paths (denoted p-rr) or nodes (denoted n-rr). In the case of paths, each agent selects at its turn its best feasible path, given the already allocated nodes (to prevent conflicts). This process operates similarly to greedy, but alternates between users to balance utilities. In the case of nodes, each agent incrementally builds the path associated with each of its graphs, by choosing in turn a next best feasible node until either the sink is reached or there is no more feasible nodes to choose (dead-end path). In the latter case, the agent is allocated the 0-utility source-to-sink path and loses the previously chosen nodes. In both approaches, constraints (1)–(6) are met since all the paths considered are feasible. Note that if there are no ties for the best path chosen by an agent at its turn, then p-rr results in a Nash equilibrium. This is not true for n-rr, since some nodes left by some agents reaching a dead-end may have prevented some other agents from finding a better solution. To overcome this difficulty, it is possible to increase the possible partial satisfaction schemes for a request, e.g., by adding arcs with a null utility from any node $v$ to the sink node.

**Example 11.** *In Figure 3, if request* a *begins the path round-robin allocation,* $\pi_{\text{p-rr}}$ *for the example in Figure 3 is equivalent to* $\pi_{\text{greedy}}$*, since* a *chooses* $\{s_a, a_3, a_2, t_a\}$ *and then* b *chooses* $\{s_b, b_1, b_4, t_b\}$*. If request* b *begins, then* b *chooses* $\{s_b, b_1, b_2, t_b\}$ *and then the only possible path for* a *is* $\{s_a, t_a\}$*, meaning that agent* A *receives a null utility.*

*If request* a *begins, the node round-robin allocation* $\pi_{\text{n-rr}}$ *is equivalent to* $\pi_{\text{greedy}}$ *because* a *first chooses* $a_3$*, then* b *chooses* $b_1$ *(only feasible option), then* a *chooses* $a_2$*, and finally* b *chooses* $b_4$ *(only feasible option). However, if request* b *begins,* b *first chooses* $b_1$*, then* a *chooses* $a_3$*, then* b *chooses* $b_2$*, and finally* a *reaches a dead-end, since the selection of* $b_2$ *implies that* $a_2$ *cannot be selected.*

## 7. Experimental Evaluation

In this section, we evaluate the different allocation methods proposed when applied to orbit slot allocation problems encoded as V-DPAP or R-DPAP. We present the experimental setup and analyze some results obtained on synthetic realistic instances. In addition to the experimental evaluation, this section also illustrates how a concrete application can be modeled in our theoretical framework.

### 7.1. Benchmarks

We first describe the benchmark generation in the case of orbit slot allocation problems.

#### 7.1.1. Constellation and Requests Features

We consider a low-Earth-orbit constellation (500 km altitude) composed of $n_p$ regularly spaced orbital planes having a 40-degree inclination, with $n_p \in \{2, 4, 8, 16\}$ and two regularly spaced satellites over each orbital plane (Walker constellation). We randomly generate requests for four agents wishing to obtain orbit slot ownerships to implement some repetitive ground acquisitions of POIs belonging to the same area. POIs are randomly selected within an extracted subset from [25], around Grenoble, France. All the agents have the same template for each request $r$, that is, communicating and getting observations every day at three requested times (RTs): $8{:}00 + \delta_r$, $12{:}00 + \delta_r$, and $16{:}00 + \delta_r$, where $\delta_r$ is uniform random time shift in $[-2h, 2h]$. Note that $\delta_r$ applies to all RTs of the same request. For each request $r$ and each RT $t$ for $r$, the slots over which orbit ownership can be claimed for achieving $r$ around time $t$ are determined thanks to a space mechanics toolbox, based on the assumption that a satellite is relevant for a POI as soon as its elevation above the horizon is greater than 15 degrees. Depending on the number of satellites in the constellation, there might not be a satellite passing over a POI exactly at the RT. We consider a tolerance window $\Delta$ equal to 1 h before and after each RT, meaning that an orbit slot is considered as valid for an RT $t$ if the middle of its temporal window is less than an hour from that RT. Finally, we impose a minimum duration *minD* of 120 s for all requests and do not consider orbit slots whose duration is shorter than this duration.

Note that these features were validated as realistic by a satellite constellation manager we work with. In fact, in the case of orbit slot allocation problems, the number of users that can afford to own orbit slots is quite low and so is their number of requests.

#### 7.1.2. From Requests to DAGs

In order to encode the problem within the DPAP framework, we first create an agent $u$ for each user that has an observation request. Then, for each request $r$ associated with agent $u$, we first create a graph $g_r$ and define a function $\mu$ such that $\mu(g_r) = u$. In a graph $g_r$ created for request $r$, the nodes are the orbit slots usable for capturing the POI targeted by $r$ at some RT, and the edges link two such consecutive orbit slots. We also add a source node that precedes all of the orbit slots of the earliest RT and a sink node that follows all of the orbit slots of the latest RT. Consequently, a path in the graph (i.e., a sequence of consecutive orbit slots) represents a way to satisfy $r$. Figure 8 represents four requests (a, b, c, and d) from four users (respectively, A, B, C, and D), with three RTs that are time 3, time 9, and time 21. In this example, each RT has at most two possible orbit slots per request. For instance, for request a, there are two orbit

slots for RT 3 ($a_1$ and $a_3$), one orbit slot for RT 9 ($a_2$), and two orbit slots for RT 21 ($a_4$ and $a_5$). Requests c and d do not have any orbit slot for RTs 3 and 9 but two each for RT 21.

For simplicity, even if the incoming arcs in the graphs of a DPAP for a given slot can have different weights, we only consider in our experiments utilities attached to the slots and not to the transitions between slots. As illustrated in Figure 9, for each candidate orbit slot for a given RT, we consider a utility function that is piecewise linear in the distance between the middle $\tau$ of the slot and that RT. The utility linearly decreases from 1 when $\tau$ is exactly on the RT to 0.25 when $\tau$ reaches the bounds of the tolerance window, i.e., RT $+\Delta$ and RT $-\Delta$. We normalize each utility with respect to the maximum utility that can be achieved for each user individually along by using its best paths. Therefore, each user's set of best paths has a utility equal to 1.

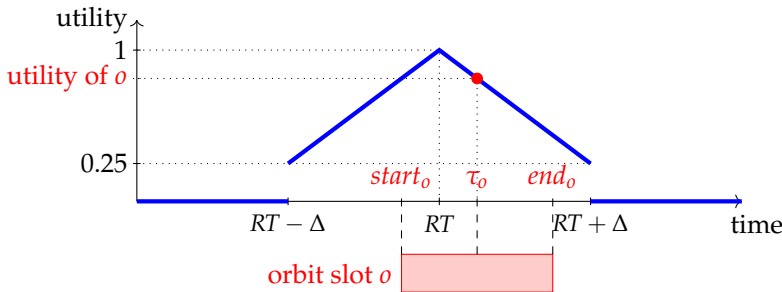

**Figure 9.** Utility function used to compute the utility of an orbit slot with respect to some RT and a tolerance $\Delta$.

In order to limit the number of edges in graphs, we add a virtual node between all slots of one RT and all slots of the next RT. If there are $n$ orbit slots for an RT $t$ and $m$ slots for the next RT $t'$, this allows there to be $n + m$ edges ($n$ edges with utility 0 going into the virtual node and $m$ edges weighted by the utility of orbit slots going out of the virtual node) instead of $n \cdot m$ edges (all $n$ nodes connected to all $m$ nodes).

Last, we consider two variants of the problem depending on whether requests can be partially satisfied.

- In the *full satisfaction* variant, each path goes through one orbit slot for each RT, except for a specific direct source-to-sink path that allows us to guarantee that there exists at least one feasible path for each request. In other words, it is not possible to skip one RT for an observation request, unless this request is not served at all.
- In the *partial satisfaction* variant, it is possible to skip some RTs for a request. In terms of generated graphs, it simply consists in adding edges with a null utility between successive virtual nodes, between the source and the first virtual node and between the last virtual node and the sink.

Figure 10 illustrates the request $a$ of Figure 8 in a full configuration (only black edges) and in a partial configuration (black and thick blue edges).

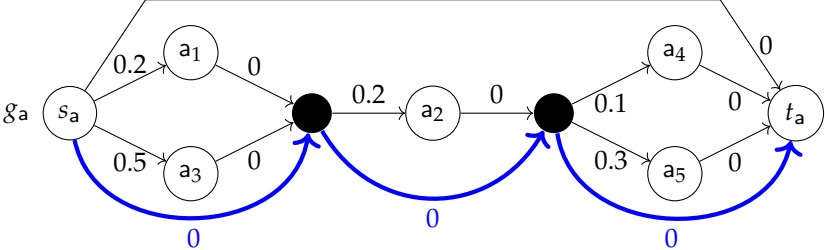

**Figure 10.** Graph for request a of Figure 8 with virtual nodes between successive RTs. The graph with only black edges represents the problem in full satisfaction mode. The graph with both black and thick blue edges represents the problem in partial request satisfaction mode.

### 7.1.3. V-DPAP and R-DPAP Generation

We describe here how the function $\phi$ is implemented in the case of the orbit slot problem for generating V-DPAP and R-DPAP instances.

V-DPAP.　For generating the set of conflicts $\mathcal{C}$ associated with a V-DPAP, we define a conflict for each pair of nodes corresponding to orbit slots that: 1. belong to the same satellite; 2. temporally overlap; and 3. are from different users. For the last assumption, we consider that it is possible to allocate to some user two orbit slots from the same satellite that overlap. In fact, as the allocation of an orbit slot consists in allowing an agent to dispose of the satellite during the associated time interval, two overlapping orbit slots $o_1$ and $o_2$ can be seen as a unique orbit slot that is the union of $o_1$ and $o_2$. With this conflict generation scheme, all the conflicts obtained are binary. Note that it would be possible to compute these conflicts more finely, for instance, by following the approach proposed in [26].

R-DPAP.　In the case of the $\phi$ function in R-DPAP, we follow the same process as in Example 4. More precisely, we create a resource $\rho_s$ for each satellite $s$ of the constellation. Then, for each graph $g_r$ associated with an observation request $r$, for each vertex $v$ in $V_{g_r}$ that corresponds to an orbit slot $o$ (i.e., all vertices except source, sink, and the ones added between successive RTs), we define $ws_{g_r}(v) = start_o$, $we_{g_r}(v) = end_o$ (i.e., the temporal window associated with vertex $v$ is exactly the temporal window associated with orbit slot $o$), $d_{g_r}(v) = minD$ (i.e., the duration associated with $v$ is the minimum duration required) and $c_{g_r}(v) = \rho_s$ where $s$ is the satellite associated with orbit slot $o$. For each vertex $v$ that is a virtual node, we consider that $ws_{g_r}(v) = we_{g_r}(v) = 0$, $d_{g_r}(v) = 0$, and $c_{g_r}(v) = r_\varnothing$. Note that R-DPAPs are next transformed into V-DPAPs as explained in Section 5.3, and in this case the conflicts obtained are not necessarily binary ones.

### 7.1.4. Instance Generation Parameters and Properties

Table 1 summarizes all the parameters used for configuring the instances. Some of these parameters do not vary, e.g., the number of requests for each user, which is equal to two. The parameters that have different values as per configuration are the number of orbital planes $n_p$, the type of problem (V-DPAP or R-DPAP), and the request mode satisfaction (full or partial). For each configuration, 100 random instances have been generated. For 2 requests per agent, 3 RTs per day, and a horizon $h = 180$ days, the DAGs generated contain $3 \cdot (2h - 1) = 1077$ layers. These settings generate DAGs having the features displayed in Table 2.

### 7.1.5. Experimental Conditions

Our experimental environment has been implemented in Java 1.8 and executed on a 20-core Intel(R) Xeon(R) CPU E5-2660 v3 @ 2.60 GHz, 62 GB RAM, Ubuntu 18.04.5 LTS. Utilitarian, leximin, and approximate leximin make use of the Java API of IBM CPLEX 20.1 (using a 2 min timeout). Note that the computation time does not need to be as tight as in Earth observation scheduling problems. In fact, in EOSPs, it might be operationally required to generate a schedule within a few minutes. Such operational constraints are not relevant for orbit slot allocation problems as plans are computed months in advance. Nevertheless, we limit the time taken by each call to the MILP solver.

**Table 1.** Generation parameters along with their possible values for configuring instances.

| Generation Parameters | | Values |
|---|---|---|
| Constellation | Altitude | 500 km |
| | Number of orbital planes $n_p$ | 2, 4, 8, 16 |
| | Number of satellites/plane | 2 |
| | Inclination | 40° |
| Scheduling horizon | Start | 1 January 2020 |
| | Duration | 180 days |
| Problems | Number of users | 4 |
| | Type | V-DPAP, R-DPAP |
| Requests | Number of requests/user | 2 |
| | Requested observation Times | 3 RTs/request |
| | Maximum random time shift $\delta_r$ | 1 h |
| | Tolerance $\Delta$ | 1 h |
| | Minimum slot duration $minD$ | 120 s |
| | Satisfaction mode | full, partial |
| Algorithms | Type | util, lex, a-lex, greedy, p-rr, n-rr |
| | CPLEX time limit | 120 s |

**Table 2.** Properties of generated problems used in the experimental evaluation (average values over 100 instances per configuration are reported).

| Problem | Properties | $n_p$ | | | |
|---|---|---|---|---|---|
| | | **2** | **4** | **8** | **16** |
| V-DPAP | Conflicts | 37,715.34 | 74,009.12 | 146,657.94 | 291,831.52 |
| | Conflict size | 2.0 | 2.0 | 2.0 | 2.0 |
| | Slots per RT | 1.94 | 3.81 | 7.54 | 15.01 |
| | Slot duration (s) | 618.10 | 616.44 | 616.91 | 616.66 |
| R-DPAP | Conflicts | 1715.38 | 3527.42 | 6981.19 | 13,929.55 |
| | Conflict size | 3.28 | 3.17 | 3.21 | 3.19 |
| | Slots per RT | 1.94 | 3.81 | 7.54 | 15.01 |
| | Slot duration (s) | 618.10 | 616.44 | 616.91 | 616.66 |

　　For each pair (problem type, request satisfaction mode) in {V-DPAP, R-DPAP} × {full, partial}, we have generated four types of plots. The first and second types of plots (e.g., Figure 11a,b) allow visualization of the average normalized global utility and the average global reward (i.e., utility not normalized), respectively, both with [0.05, 0.95] as a confidence interval [2] for each constellation size and for each algorithm. In the second type of plot (e.g., Figure 11c), the average computation time (logarithmic time scale) is indicated, also for each constellation size and each algorithm. Finally, the fourth type of plot (e.g., Figures 12) allows us to analyze the fairness of the resulting allocations. More precisely, we show the average utility profile in all instances for each algorithm and for each constellation size. Such a utility profile is in leximin order: for each radar, among the four agents, the south represents the agent having the best utility over all agents, the west is the second best utility, the north is the third best utility, and the east corresponds to the agent with the worst utility. For some cases, we sometimes detail utility profiles obtained by each algorithm in some specific instances.

　　We first present results associated with the full request satisfaction mode, and then results associated with the partial request satisfaction mode.

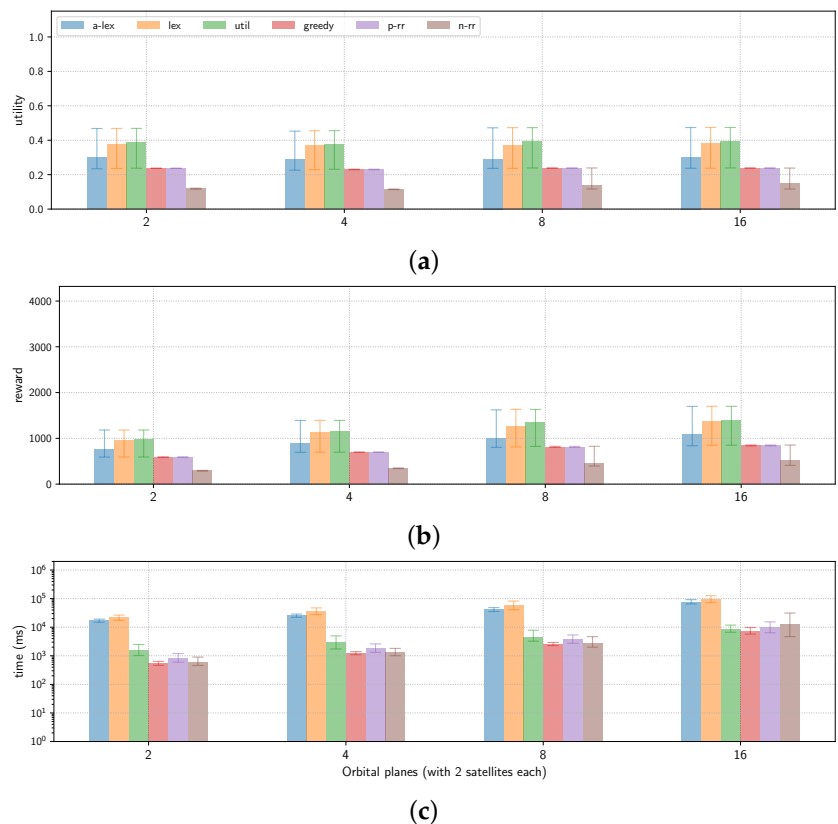

(**a**)

(**b**)

(**c**)

**Figure 11.** Performance metrics obtained by each algorithm for each constellation size, for full request satisfaction mode and encoded as V-DPAP. (**a**) Normalized utility; (**b**) global reward; (**c**) computation time.

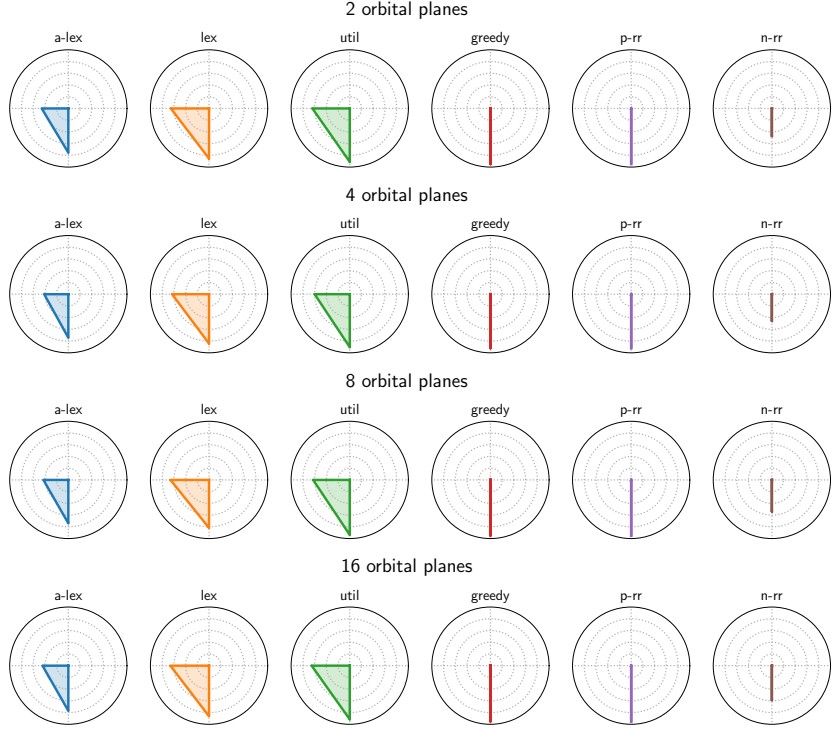

**Figure 12.** Average utility profiles (in leximin order) for each constellation size and each algorithm (south: best utility over all agents; west: second best utility; north: third best utility; east: worst utility), for full request satisfaction mode and encoded as V-DPAP.

### 7.2. Results for the Full Request Satisfaction Mode

#### 7.2.1. V-DPAP Results Analysis for $n_p = 2$

Figure 11a compares the normalized utility obtained by each algorithm. As expected, the utilitarian allocation algorithm (util) returns the best global utility. Such a utility is nevertheless quite low, as it does not reach 0.4 on average. The leximin allocation algorithm (lex) is the second in terms of the normalized utility. Its normalized utility score is slightly lower than that of the util. The approximate leximin (a-lex) algorithm's utility is around 0.3. The fact that a-lex's utility is lower than lex's comes from the fact that a-lex cannot backtrack on its decision on the agent's order in the leximin vector, which is prejudicial in the case of the utility's equality within agents. Greedy allocation (greedy) and path round-robin allocation (p-rr) have almost the same global utility (around 0.2) and finally, node round-robin allocation (n-rr) has a global utility lower than 0.1. In terms of global reward, this corresponds to 1000 for the best algorithm (util) and around 150 for the worst one (n-rr).

The time required by each algorithm is reported in Figure 11c. The most time-consuming approaches are a-lex and lex (around 20 s). In fact, they have to call the MILP solver (CPLEX) as many times as the number of agents (here four). Algorithms greedy, n-rr, and p-rr are the fastest ones as they return a solution in less than a second. Algorithm util returns a solution in approximately 10 s.

The top line of Figure 12 displays the average utility profiles involving two orbital planes. The best served agent has a utility very close to 1. Such radars show that the worst served and second worst served agents all have a null utility. This comes from the fact that the corresponding instances are very conflicting. Once two agents receive a path with a utility strictly greater than 0, this prevents the others from satisfying their requests. We can notice that algorithms util and lex have very similar utility profiles on average. Algorithm a-lex does not perform as well for fairness, specifically for the second best served agent. Algorithms n-rr, p-rr, and greedy serve only one agent.

#### 7.2.2. Sensitivity to Constellation Size

The comparison between the algorithms for utility, computation time, and leximin profiles does not change with respect to the number of orbital planes. In other words, the relative performance of the algorithms is the same whatever the size of the constellation. Figure 11a shows that the normalized global utility obtained by the agents does not increase a lot with the constellation's size. However, the allocation's global reward increases with the growing number of orbital planes. In fact, as shown in Figure 11b, the global reward obtained for 2 orbital planes (i.e., a constellation with 4 satellites) is around 1000 for the util algorithm. When considering 16 orbital planes (i.e., a constellation with 32 satellites), such a reward almost reaches 1500, at best. The fact that the normalized utility does not increase with the constellation size but the global reward does, comes from the normalization factor. In fact, with 32 satellites, the global utility that the agents can obtain individually is higher than with 4 satellites. However, the global utility obtained by the agents is relatively the same compared with their best paths and results in a similar normalized global utility.

The time required for computing the allocations also increases with the size of the constellation. More precisely, Figure 11c shows that computation time is multiplied by 10 when the number of satellites increases from 4 to 32. This comes from the higher number of orbit slots and consequently much larger graphs (see Table 2) with more paths to explore and more constraints to check.

The average utility profiles given in Figure 12 show that the utility profiles obtained by algorithms do not change much with the constellation size. Even with more satellites, at the most two agents have a utility strictly greater than 0. This confirms the high number of conflicts of the requests in the considered instances. This illustrates the low utility and reward obtained in this setting: few requests are fulfilled in the end.

These results show that, in the case of V-DPAP with full request satisfaction mode, algorithm util is quite interesting in terms of global utility versus required time. Moreover,

as the instances do not allow the utility profiles to be balanced between the agents, this algorithm provides as fair allocations as algorithm lex.

### 7.2.3. R-DPAP Results

Figures 13–15 present the results associated with R-DPAP instances in the full request mode satisfaction. The algorithms behave relatively to each other as for the V-DPAP case. More precisely, with respect to the global utility, algorithm util returns the best global utility, then, lex, a-lex, greedy, and n-rr equivalently, and finally, n-rr.

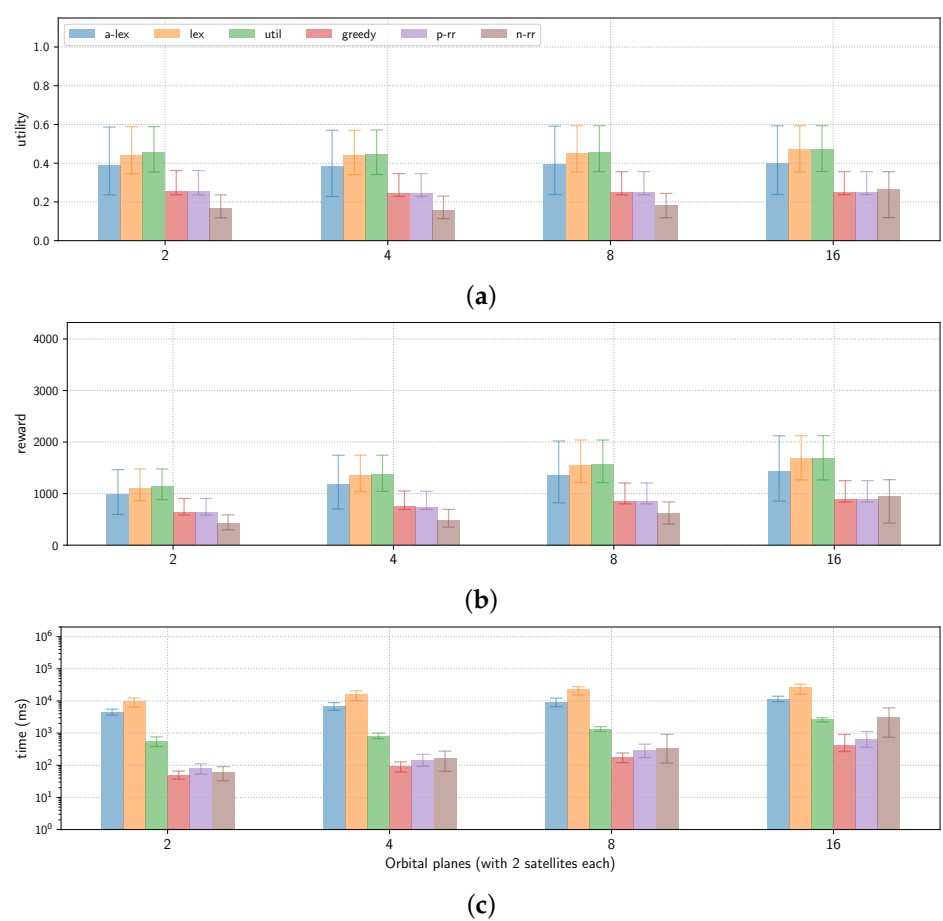

**Figure 13.** Performance metrics obtained by each algorithm for each constellation size, for full request satisfaction mode and encoded as R-DPAP. (**a**) Normalized utility; (**b**) global reward; (**c**) computation time.

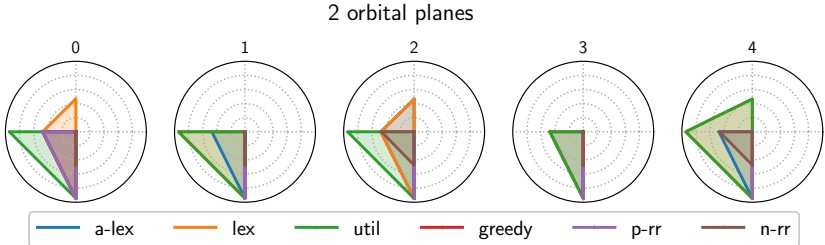

**Figure 14.** Utility profiles (in leximin order) for the first 5 instances for a constellation with 2 orbital planes (4 satellites) and each algorithm (south: best utility over all agents; west: second best utility; north: third best utility; east: worst utility), for full request satisfaction mode and encoded as R-DPAP.

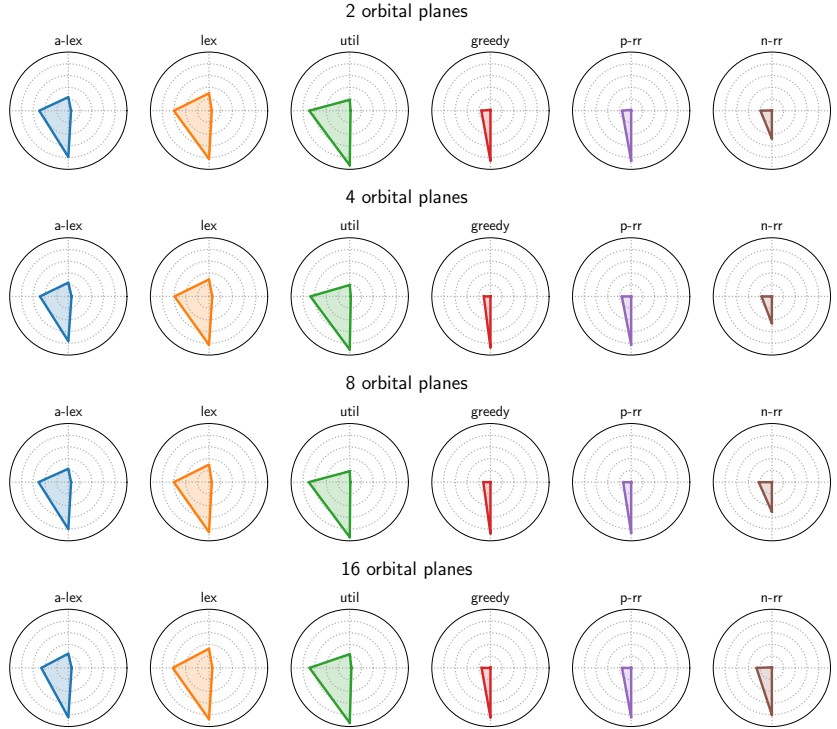

**Figure 15.** Average utility profiles (in leximin order) for each constellation size and each algorithm (south: best utility over all agents; west: second best utility; north: third best utility; east: worst utility), for full request satisfaction mode and encoded as R-DPAP.

In comparison with the V-DPAP results, instances encoded in the R-DPAP framework allow a higher utility to be reached. First, the normalized global utility (Figure 13a) is around 0.5 for algorithm util with 32 satellites. Note that it is only a little less with 4 satellites (around 0.45). In terms of global reward (Figure 13b), the average score is around 1700 for util with 32 satellites.

The fact that the utility is higher in the R-DPAP benchmark than in the V-DPAP framework comes from the fact that the first allows the orbit slots to be split while the second does not. Consequently, with R-DPAP, when an agent is given an orbit slot on a path with a non-null utility, overlapping orbit slots for other agents might still be selectable. Such a phenomenon can be confirmed by the radars of Figure 14. Indeed, instance 0 with 2 orbital planes (left radar) shows that it is possible for three agents out of four to have a non-null utility. Such an allocation is obtained with algorithm lex. In this case, note that algorithm a-lex performs worse than lex in the sense that two agents have a zero utility with a-lex. This is probably due to the fact that both algorithms compute that the worst served agent has a null utility, but a-lex has to choose to which agent this null utility is allocated. In the case of a bad choice, this prevents a-lex from obtaining a higher utility for the second worst served agent.

The left radar of Figure 14 also shows that util tends to favor agents with high utilities (two agents with utility equal to 1, and two agents with 0), whereas lex splits utility between agents (best agent with utility 1, two others with utility 0.45, and the last with 0). The average utility profiles of Figure 15 confirm this difference of behavior between algorithms util and lex. As for V-DPAP, algorithm a-lex's performance is lower than util and lex with respect to fairness. Other approaches manage on average to serve a second agent but with a very low utility.

Finally, the order of magnitude for the time required to compute solutions is the same between V-DPAP and R-DPAP.

These results show that, in the case of R-DPAP with full request mode satisfaction, the best trade-off between global utility and computation time is given by algorithm util.

However, in terms of fairness, this algorithm is not as good as algorithm lex in several instances, even if lex gives larger computation times.

Note that R-DPAP is still parametric in the sense that it requires defining the duration *minD* (here 120 s) requested in each orbit slot. With a low *minD* value, orbit sharing can be possible, while using a high *minD* value may prevent such splitting, and in the extreme case R-DPAP becomes equivalent to V-DPAP, utility-wise.

### 7.3. Results for the Partial Request Satisfaction Mode

We now analyze the results for the instances in which requests can be partially satisfied by skipping some RTs.

### 7.3.1. V-DPAP Results

Figures 16 and 17 show the results for instances encoded as V-DPAP. From Figure 16a, we can observe that the normalized utility is much higher than with instances encoded in V-DPAP with the request full satisfaction mode. For instance, for a constellation involving 4 satellites, algorithms util, lex and a-lex almost reach a 0.6 normalized utility value. For 32 satellites, this normalized utility is equal to 0.85. In terms of reward (Figure 16b), the global reward is also much higher. Note that the relative performance of the algorithms is the same as for V-DPAP with the full satisfaction mode, i.e., algorithm util returns the allocation with the best global utility, then lex, a-lex, p-rr, greedy, and n-rr. Nevertheless, with 32 satellites, all algorithms except n-rr return allocations with approximately the same global utility. This increase in performance with the change in request mode satisfaction shows that even if paths conflict, the skip possibility allows many more requests to be tackled.

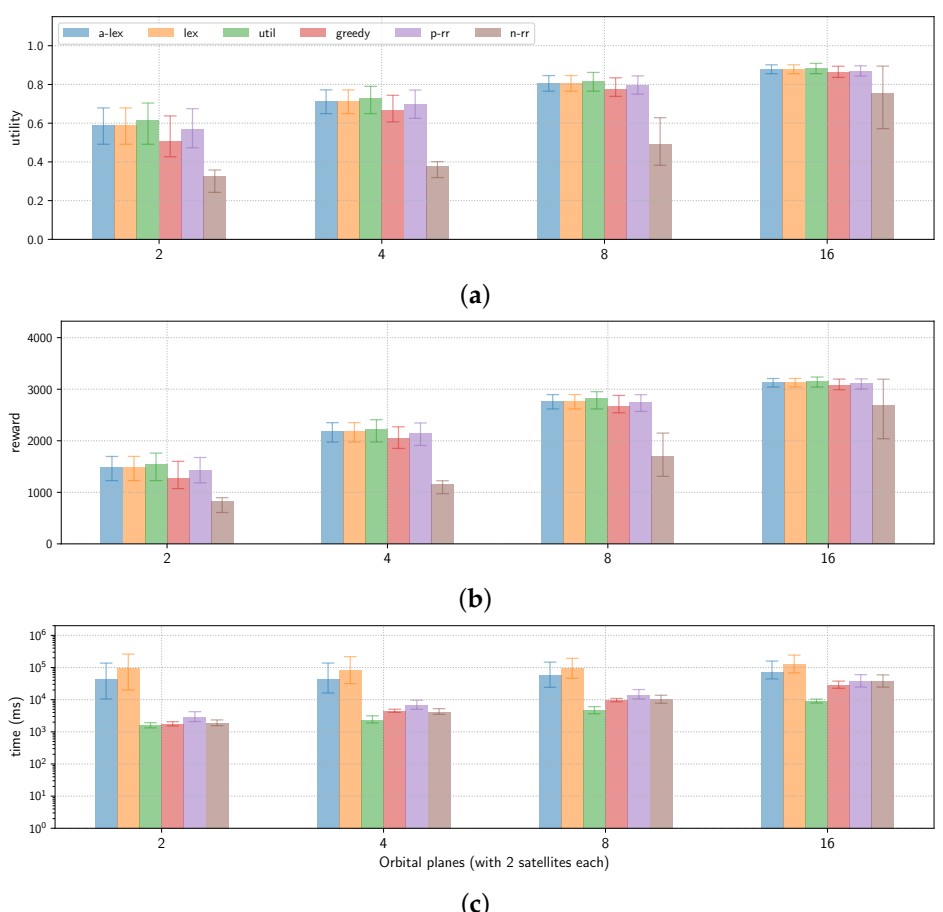

**Figure 16.** Performance metrics obtained by each algorithm for each constellation size, for flexible requests encoded as V-DPAP. (**a**) Normalized utility; (**b**) global reward; (**c**) computation time.

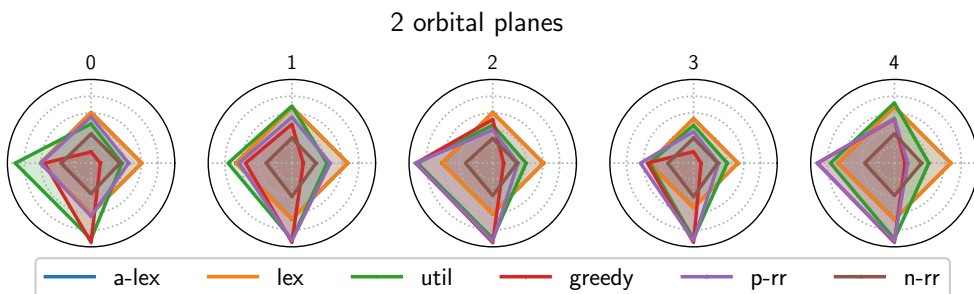

**Figure 17.** Utility profiles (in leximin order) for the first 5 instances for a constellation with 2 orbital plans (4 satellites) and each algorithm (south: best utility over all agents; west: second best utility; north: third best utility; east: worst utility), for flexible requests encoded as V-DPAP.

From Figure 16c, we can first notice that the time required by algorithms lex and a-lex is much higher than for V-DPAP with request full satisfaction mode. In fact, for algorithm lex, 10 s are required for V-DPAP with request full satisfaction mode but 100 for V-DPAP with request partial satisfaction mode. However, for these algorithms, the order of magnitude does not change with the constellation size. Such a phenomenon is probably due to the fact that there are a lot of complex paths (i.e., paths that are not *source → sink*) with the same utility, which makes it harder to compute the worst utility for a given agent. Algorithms greedy, p-rr, and n-rr also require much more time than for instances in V-DPAP with full request satisfaction mode. This can be explained by the fact the number of paths is much larger but that nodes still belong to several conflicts. Therefore, every time a path is selected in a graph, other graphs have many nodes that are deactivated, which forces new best paths to be computed and overall requires some computation time. In comparison, algorithm util requires approximately the same time in the partial and full satisfaction modes.

Next, Figures 17 and 18 show that in the partial satisfaction mode, the utility profiles are much more balanced between agents. The radars in Figure 17 allow the algorithms' behaviors to be compared over some instances involving two satellites. It shows that algorithm util favors high utilities, which is sometimes quite fair (instance 3) and sometimes not (instance 0). Algorithm greedy serves very well one agent but cannot serve well the others because of conflicts between paths. Algorithm p-rr performs a little better than greedy in terms of fairness. Algorithms lex and a-lex allow the utility to be balanced between the agents. For instance, the top line radars show that it is possible to reach a solution where all agents have approximately the same utility (around 0.6). Algorithm n-rr is also quite fair but the utility per agent is much lower (0.2). Figure 18 shows that these comments can be generalized to all instances on average.

In the case of a larger constellation, the algorithms (except n-rr) behave almost the same in terms of leximin vectors, and there exist solutions where all agents can be served quite well.

These results show that in the case of V-DPAP with requests partial satisfaction mode, algorithm util offers the best utility/time trade-off. However, in terms of fairness, such an algorithm gives good performance only for constellations with at least 8 orbital planes (16 satellites). For a smaller number of satellites, algorithms lex and a-lex can be much fairer (depending on the instance), despite a greater computation time.

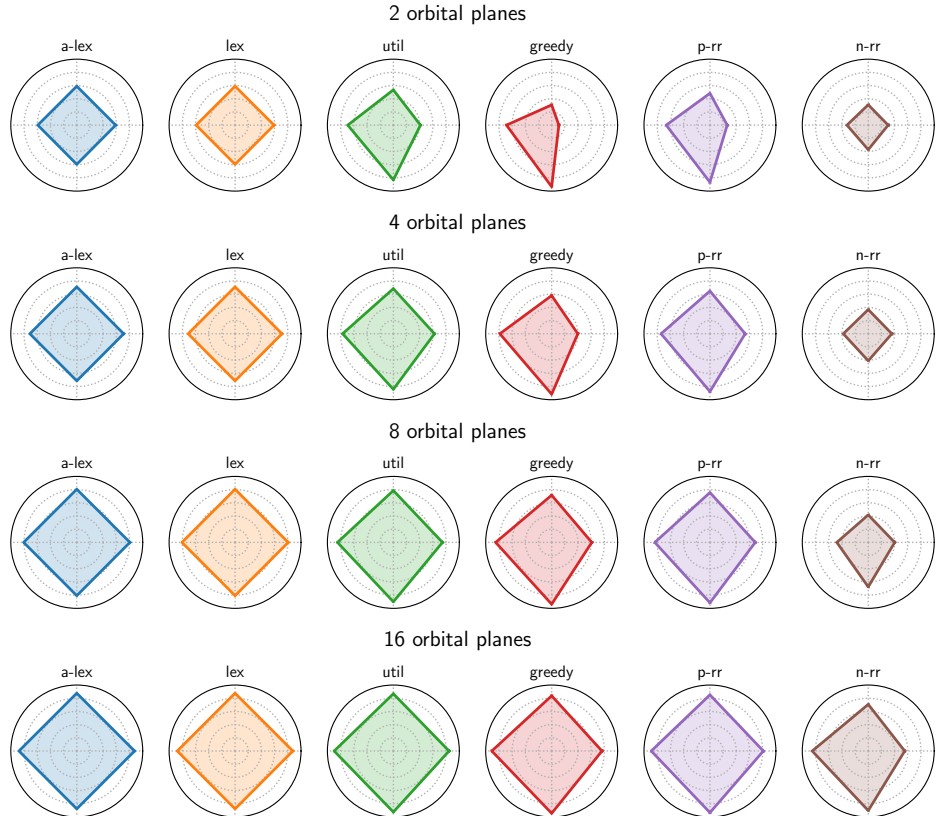

**Figure 18.** Average utility profiles (in leximin order) for each constellation size and each algorithm (south: best utility over all agents; west: second best utility; north: third best utility; east: worst utility), for partial request satisfaction mode and encoded as V-DPAP.

### 7.3.2. R-DPAP Results

In the case of R-DPAP with partial request satisfaction mode, Figure 19a shows that the maximum utility is reached by all algorithms whatever the constellation size, except for n-rr. Note that the obtained global normalized utility is not equal to 1 because there are still some conflicts between some orbit slots that prevent the agents from obtaining their best paths.

Figure 19b shows that the global reward increases with the number of satellites in the constellation. In fact, the larger the constellations, the higher the number of orbit slots and the higher the number of paths with a higher utility in the graphs.

For all of the algorithms, the computation time required is much lower than for V-DPAP with partial satisfaction mode. This is quite natural, since even if there is a large number of paths, the selection of one path for an agent does not require deactivating many nodes in other graphs. This comes from the fact that orbit slots can be split between agents, which results in less conflicts between nodes.

We do not provide here radars per instance, since the profiles obtained by the algorithms all overlap. Indeed, Figure 20 confirms that all the agents have a utility almost equal to 1 for all the algorithms except n-rr. The latter struggles with highly conflicting settings. With less conflicting settings (with more satellites) n-rr drastically improves its performance, since there is less chance to reach a situation where an agent must skip one RT.

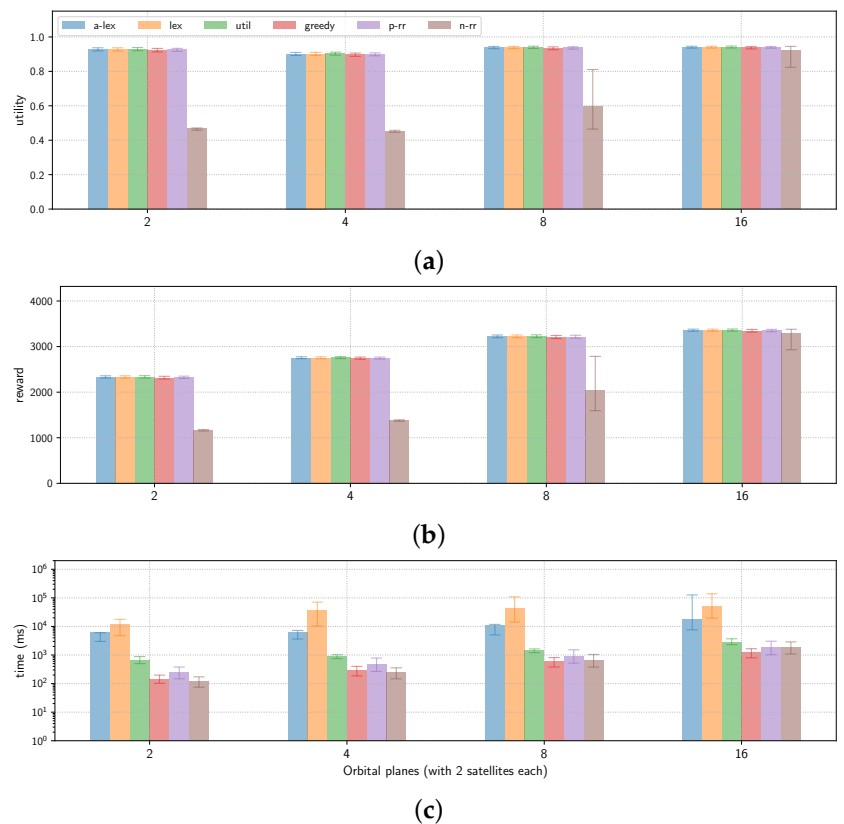

(a)

(b)

(c)

**Figure 19.** Performance metrics obtained by each algorithm for each constellation size, for flexible requests encoded as R-DPAP. (**a**) Normalized utility; (**b**) global reward; (**c**) computation time.

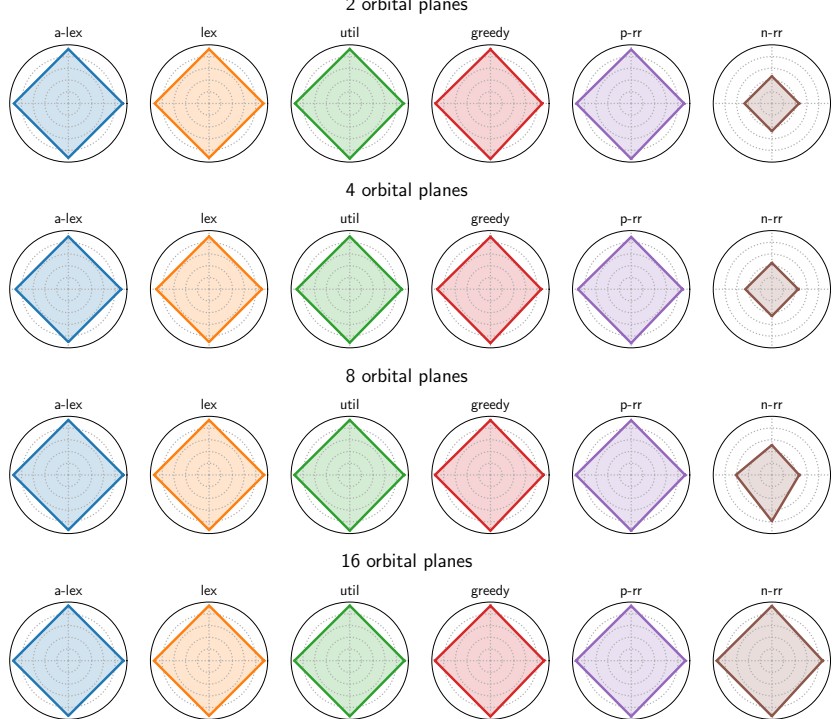

**Figure 20.** Average utility profiles (in leximin order) for each constellation size and each algorithm (south: best utility over all agents; west: second best utility; north: third best utility; east: worst utility), for partial request satisfaction mode and encoded as R-DPAP.

These results suggest that in the case of R-DPAP with full request satisfaction mode, algorithm greedy offers the best trade-off in quality/time as it allows a fair allocation to be reached with a high global utility (as for other algorithms) but in much less time.

## 8. Conclusions

In this paper, we proposed several models for novel resource allocation problems where agents express their preferences over conflicting bundles of items as edge-weighted DAGs (DPAP). We particularly focused on conflicts on vertices (V-DPAP) and conflicts on resources (R-DPAP). We introduced and analyzed several solution methods (utilitarian, leximin, approximate leximin, greedy) against the classically used round-robin allocations from the utilitarianism and fairness perspectives. We evaluated these methods on large randomly generated instances of orbit slot allocation problems, where requests could be fully or partially fulfilled. We showed that when requests must be fully fulfilled, allowing resource sharing via R-DPAP encoding improves the performance of the system compared to V-DPAP with respect to normalized utility and global reward, while the computation times are equivalent or lower. When considering the request full satisfaction mode, problems encoded as V-DPAP are much more constrained with respect to the number of agents that can receive a non-empty allocation. Therefore, algorithm util is a relevant approach. In the case of R-DPAP, algorithm a-lex provides good results with respect to utility and is much fairer than other approaches, even if it requires a longer computation time. In the case of partial request satisfaction mode and V-DPAP problems, there is no clear winner on all metrics for small constellations: lex clearly returns fair allocations with a good global utility but requires a long computation time. On the other hand, algorithm util is faster but not as fair. For large constellations, algorithm util allows us to reach the fairest allocations and is, therefore, the most suitable. Finally, when offering even more flexibility, i.e., allowing partial request fulfilling, the performances become even better, to a point where, for larger constellations, all the algorithms reach the same optimal and fair allocations. This highlights that adding request flexibility eases the allocation process, whilst the problems remain NP-hard in general. In such a case, non-exact algorithms such as greedy offer the best trade-off with respect to utility, fairness, and computation time.

We identify several tracks for future investigations. First, as DPAPs are strongly constrained by conflicts, we aim to explore minimum conflict heuristics to improve our algorithms. Secondly, we believe DPAP and its variants have great potential to be used in a variety of domains, and we thus aim to evaluate the proposed techniques on problems coming from other application fields, such as the NFV domain (function chains modeled as graphs and incompatibilities controlling the access to nodes) or the multi-agent path finding domain (path preferences modeled as graphs and incompatibilities, imposing that two agents cannot occupy the same position at the same time). Depending on the targeted application, other ways for expressing conflicting bundles could be explored. For instance, one could consider that items can consume resources with capacity. Finally, in the Earth observation domain, once the slots have been allocated, the agents have to plan their own observations within the allocated slots, and may have to interact to accept external observations. Such a coordination scheme has been investigated [3], but we aim to evaluate the whole chain (slot allocation followed by coordinated observation scheduling) on realistic data.

**Author Contributions:** Conceptualization, S.R., G.P., C.P. and S.M.; data curation, S.R., G.P., C.P. and S.M.; formal analysis, S.R., G.P. and C.P.; funding acquisition, S.R. and C.P.; investigation, S.R., G.P., C.P. and S.M.; methodology, S.R., G.P., C.P. and S.M.; project administration, S.R., G.P. and C.P.; resources, S.R., G.P., C.P. and S.M.; software, S.R., G.P., C.P. and S.M.; supervision, S.R., G.P., C.P. and S.M.; validation, S.R., G.P., C.P. and S.M.; visualization, S.R., G.P. and C.P.; writing—original draft, S.R., G.P. and C.P.; writing—review and editing, S.R., G.P., C.P. and S.M. All authors have read and agreed to the published version of the manuscript.

**Funding:** This work has been performed with the support of the French government in the context of the "Programme d'Invertissements d'Avenir", namely, by the BPI PSPC LiChIE project (Lion Chaine Image Elargie), coordinated by Airbus Defence and Space.

**Data Availability Statement:** Instances used in the experimental evaluation are available for the community, and accessible on Zenodo (https://doi.org/10.5281/zenodo.7669379).

**Conflicts of Interest:** The authors declare no conflict of interest.

## Abbreviations

The following abbreviations are used in this manuscript:

| | |
|---|---|
| DAG | Directed acyclic graph |
| DPAP | Directed path allocation problem |
| V-DPAP | Vertex-constrained DPAP |
| R-DPAP | Resource-constrained DPAP |
| PADAG | Path allocation in directed acyclic graph |
| ILP | Integer linear programming |
| MILP | Mixed integer linear programming |
| POI | Point of interest |
| RT | Request time |
| lex | Leximin solver |
| a-lex | Approximate leximin solver |
| greedy | Greedy solver |
| util | Utilitarian MILP solver |
| p-rr | Path round-robin solver |
| n-rr | Node round-robin solver |

## Notes

| | |
|---|---|
| 1 | A Nash equilibrium is an allocation in which the modification of a path for a single agent does not improve its associated utility. |
| 2 | We have removed the worst 5% of values and the best 5% of values for the indicated range. |

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
