# Peer review of "Conflicting Bundle Allocation with Preferences in Weighted Directed Acyclic Graphs: Application to Orbit Slot Allocation Problems†"

_systems, doi:10.3390/systems11060297_

Round 1

Reviewer 1 Report

 This paper introduce resource allocation techniques for problems where the agents express requests for obtaining item bundles as compact edge-weighted directed acyclic graphs. The goal of the problem is to allocate a path in the DAG for each client, satisfying the vertex compatibility constraints and resource compatibility constraints, respectively. In my opinion, the topic of this study is interesting and valuable in practice, and the contributions are sufficient. However, I still have the following comments for this paper.

1) In this paper, a path in a DAG was allocated to a client, which is similar to the traditional satellite scheduling problem, and the difference is that the conflicts  between paths need to be considered. However, I wonder how to judge and compute whether two path are conflicted. In addition, as we know, the number of the paths in a DAG is exponential, how to tackle the  exponential explosion of the number of the paths.

2) The differences between the Orbit Slot Allocation Problem and the traditional EOS scheduling should be more highlighted and described more detaildly.

3) Related works should be enriched and revised, the studies of [4] was not clearly described, and the difference and conttibutions compared to [4] and [16] was not highlighted. More references related to satellite and orbit allocation should be included, such as:

[1] Eddy, D. , and  M. J. Kochenderfer . "A Maximum Independent Set Method for Scheduling Earth-Observing Satellite Constellations." Journal of Spacecraft and Rockets 5(2021).

[2] Jianjiang Wang, Guopeng Song, Zhe Liang, et al. Unrelated parallel machine scheduling with multiple time windows: An application to earth observation satellite scheduling. Computers & Operations Research, 2023, 149: 106010.

4) The English writting could be improved, such as page 1, line 19, "satellite performs its images", Page 2, line 66,"path allocation problem with conflict (Directed Path Allocation Problem, or DPAP) along with two optimization criteria", etc.

Author Response

We thank the reviewer for their comments on this paper. 

In the following, we try to answer the comments. We attach a pdf version of the revised manuscript that includes new elements related to those comments. 

1) In this paper, a path in a DAG was allocated to a client, which is similar to the traditional satellite scheduling problem, and the difference is that the conflicts  between paths need to be considered. However, I wonder how to judge and compute whether two path are conflicted. In addition, as we know, the number of the paths in a DAG is exponential, how to tackle the  exponential explosion of the number of the paths.

A conflict between two paths represents the fact that assigning these paths to clients is infeasible (e.g. because of some overlapping slots) or strongly undesirable for the constellation manager. 
In the general case, the definition of the compatibility function relies on a list of pairs of paths that are compatible (or incompatible) with each other. 
As indicated by Reviewer 1, the number of paths in a DAG is exponential, which makes the definition impractical. This is why we define two other definitions of compatibility functions, respectively through the VDPAP and RDPAP formulations, that allow to represent and compute conflicts. 

We have updated the associated paragraph in order to make that point clearer in the paper (lines 201-208).

2) The differences between the Orbit Slot Allocation Problem and the traditional EOS scheduling should be more highlighted and described more detaildly.

We have added a dedicated paragraph at the end of the Related Works Section (lines 142-165).

3) Related works should be enriched and revised, the studies of [4] was not clearly described, and the difference and conttibutions compared to [4] and [16] was not highlighted. More references related to satellite and orbit allocation should be included, such as:

[1] Eddy, D. , and  M. J. Kochenderfer . "A Maximum Independent Set Method for Scheduling Earth-Observing Satellite Constellations." Journal of Spacecraft and Rockets 5(2021).

[2] Jianjiang Wang, Guopeng Song, Zhe Liang, et al. Unrelated parallel machine scheduling with multiple time windows: An application to earth observation satellite scheduling. Computers & Operations Research, 2023, 149: 106010.

We have added a more detailed description of [16] in the Related Works Section. As the work done in [4] is described in detail in Section 4. We have clearly mentioned that in the Related Works Section. 

We have added references [1] and [2], as proposed by Reviewer 1.

4) The English writting could be improved, such as page 1, line 19, "satellite performs its images", Page 2, line 66,"path allocation problem with conflict (Directed Path Allocation Problem, or DPAP) along with two optimization criteria", etc.

We have taken the remarks into account and have tried to improve the English. 

Reviewer 2 Report

This paper defines a generic modeling framework for directed path allocation problem (DPAP) and shows its instances; Vertex-constrained DPAP (V-DPAP) and Resource-constrained DPAP(R-DPAP).

Furthermore, this paper proves the NP-completeness of the decision problem of V-DPAP and R-DPAP and defines algorithms for solving V-DPA and R-DPAP.

Experiments evaluations show that the algorithm efficiently solves orbit slot application problems from quality, computation time, and fairness viewpoints. 

The allocation problem has been regarded as a classical research topic recently, but this paper revives and applies it to interesting novel problems.

The paper is well-structured and logically developed.

Therefore, as a reviewer, I believe the paper can be accepted as a research article in the journal.

It would be useful for the reader to have an explanation of what the actual problem size should be.

It would be helpful for the reader to explain the reasonability of the experimental settings compared to the actual current and near-future situations.

Author Response

We thank the reviewer for their comments.

We have attached to our answer the revised manuscript that contains new elements. 

It would be useful for the reader to have an explanation of what the actual problem size should be.

> We have added a comment of the actual problem size in 7.1.1. Constellation and Requests Feature

It would be helpful for the reader to explain the reasonability of the experimental settings compared to the actual current and near-future situations.

> We have added a comment on that point in the first paragraph of 7.1.5. Experimental Conditions.

Reviewer 3 Report

1. The  motivation and the validation of this work are not clearly described. Although the proposed method and system model is explicitly provided, more details of the evaluation part should be given.

2. What is the fundamental for the utility function? Since the fig.9 provide a curve for it, it is still not clear how is it scaled. 

3. The author claims that the fairness is improved in the proposed method, but the analysis is not provided. How is faireness exactly defined in this work?

4. I think the proposed resource method can be compared with the following literatures:

[1] "Dynamic Resource Allocation with Deep Reinforcement Learning in Multibeam Satellite Communication," in IEEE Wireless Communications Letters

[2] QoS-Guarantee Resource Allocation for Multibeam Satellite Industrial Internet of Things With NOMA," in IEEE Transactions on Industrial Informatics

Author Response

We thank the reviewer for their comments. 

Please find our answers to the comments in the following. Note that we have attached the revised version of the manuscript. 

1. The  motivation and the validation of this work are not clearly described. Although the proposed method and system model is explicitly provided, more details of the evaluation part should be given.

As formally described in Section 3, after Definition 5, the problems we consider in this paper are (i) how to compute an optimal (utilitarian) valid allocation \pi that maximizes u(\pi), and (ii) how to compute an optimal fair valid allocation \pi that maximizes lex(\pi). Through the first criteria, we maximize the overall profit and through the second, we maximize the fairness. In the experiments, we stick to those two criteria and evaluate the scores obtained for each of them and for each method. Note that some of these methods are complete, which allows us to have an idea of the optimal solution (when solvers can provide it). 

Moreover, the dataset used for experiments will be made available on a public repository (Zenodo). 

2. What is the fundamental for the utility function? Since the fig.9 provide a curve for it, it is still not clear how is it scaled. 

Utility is normalized with respect to the best utility that each agent can achieve individually. This means that when an agent has a utility equal to 1, she's allocated the best path for her. We have made that point clear in the text associated to Fig. 9.

3. The author claims that the fairness is improved in the proposed method, but the analysis is not provided. How is faireness exactly defined in this work?

The fairness is modelled by the leximin vector, as formally defined in Definition 5. As classically done in several resource allocation problems works, maximizing the fairness comes to maximizing the leximin vector, which we do in this paper. 

4. I think the proposed resource method can be compared with the following literatures:

[1] "Dynamic Resource Allocation with Deep Reinforcement Learning in Multibeam Satellite Communication," in IEEE Wireless Communications Letters

[2] QoS-Guarantee Resource Allocation for Multibeam Satellite Industrial Internet of Things With NOMA," in IEEE Transactions on Industrial Informatics

While these papers focus on Resource Allocation problems in the context of satellites, the addressed problems are very different from the one we study in the paper. In fact, in those problems, communication satellites are geostationnary, which means that there are no temporal constraints linked to the visibility of Earth targets. The allocation problems consists in choosing which channel to use for a given set of target areas on Earth, and not to share spatio-temporal windows for satellites moving over their orbits. This makes this two families of problems (and related solutions) not comparable.

Reviewer 4 Report

It is an interesting resource allocation problem; description is extensive and long, but it is quite complete. You do not need to read  other papers to understand it. 
Still, it would be good if it could be shortened. 

Author Response

We would like to thank the reviewer for their comments.

Here is our answer to the main comment of the review.

It is an interesting resource allocation problem; description is extensive and long, but it is quite complete. You do not need to read  other papers to understand it. 
Still, it would be good if it could be shortened. 

In this article, as we did not have an explicit maximum number of pages, we have tried to add several examples to illustrate the concepts and to provide experimental results covering various configurations. In the end, we believe that shortening the paper would downgrade its completeness, and we would like to avoid removing materials that may have been useful for other reviewers.